# Adversarial Support Alignment

**Shangyuan Tong**[*]
MIT CSAIL

**Timur Garipov**[*]
MIT CSAIL

**Yang Zhang**
MIT-IBM Watson AI Lab

**Shiyu Chang**
UC Santa Barbara

**Tommi Jaakkola**
MIT CSAIL

## Abstract

We study the problem of aligning the supports of distributions. Compared to the existing work on distribution alignment, support alignment does not require the densities to be matched. We propose symmetric support difference as a divergence measure to quantify the mismatch between supports. We show that select discriminators (e.g. discriminator trained for Jensen–Shannon divergence) are able to map support differences as support differences in their one-dimensional output space. Following this result, our method aligns supports by minimizing a symmetrized relaxed optimal transport cost in the discriminator 1D space via an adversarial process. Furthermore, we show that our approach can be viewed as a limit of existing notions of alignment by increasing transportation assignment tolerance. We quantitatively evaluate the method across domain adaptation tasks with shifts in label distributions. Our experiments[1] show that the proposed method is more robust against these shifts than other alignment-based baselines.

## 1 Introduction

Learning tasks often involve estimating properties of distributions from samples or aligning such characteristics across domains. We can align full distributions (adversarial domain alignment), certain statistics (canonical correlation analysis), or the support of distributions (this paper). Much of the recent work has focused on full distributional alignment, for good reasons. In domain adaptation, motivated by theoretical results (Ben-David et al., 2007; 2010), a series of papers (Ajakan et al., 2014; Ganin & Lempitsky, 2015; Ganin et al., 2016; Tzeng et al., 2017; Shen et al., 2018; Pei et al., 2018; Zhao et al., 2018; Li et al., 2018a; Wang et al., 2021; Kumar et al., 2018) seek to align distributions of representations between domains, and utilize a shared classifier on the aligned representation space.

Alignment in distributions implies alignment in supports. However, when there are additional objectives/constraints to satisfy, the minimizer for a distribution alignment objective does not necessarily minimize a support alignment objective. Example in Figure 1 demonstrates the qualitative distinction between two minimizers when distribution alignment is not achievable. The distribution alignment objective prefers to keep supports unaligned even if support alignment is achievable. Recent works (Zhao et al., 2019; Li et al., 2020; Tan et al., 2020; Wu et al., 2019b; Tachet des Combes et al., 2020) have demonstrated that a shift in label distributions between source and target leads to a characterizable performance drop when the representations are forced into a distribution alignment. The error bound in Johansson et al. (2019) suggests aligning the supports of representations instead.

In this paper, we focus on distribution support as the key characteristic to align. We introduce a support divergence to measure the support mismatch and algorithms to optimize such alignment. We also position our approach in the spectrum of other alignment methods. Our contributions are as follows (all proofs can be found in Appendix A):

1. In Section 2.1, we measure the differences between supports of distributions. Building on the Hausdorff distance, we introduce a novel support divergence better suited for optimization, which we refer to as *symmetric support difference* (SSD) divergence.

---

[*]First two authors contributed equally. Correspondence to Shangyuan Tong (sytong@csail.mit.edu).
[1]We provide the code reproducing experiment results at https://github.com/timgaripov/asa.

2. In Section 2.2, we identify an important property of the discriminator trained for Jensen–Shannon divergence: support differences in the original space of interest are "preserved" as support differences in the one-dimensional discriminator output space.

3. In Section 3, we present our practical algorithm for support alignment, *Adversarial Support Alignment* (ASA). Essentially, based on the analysis presented in Section 2.2, our solution is to align supports in the discriminator 1D space, which is computationally efficient.

4. In Section 4, we place different notions of alignment – distribution alignment, relaxed distribution alignment and support alignment – within a coherent spectrum from the point of view of optimal transport, characterizing their relationships, both theoretically in terms of their objectives and practically in terms of their algorithms.

5. In Section 5, we demonstrate the effectiveness of support alignment in practice for domain adaptation setting. Compared to other alignment-based baselines, our proposed method is more robust against shifts in label distributions.

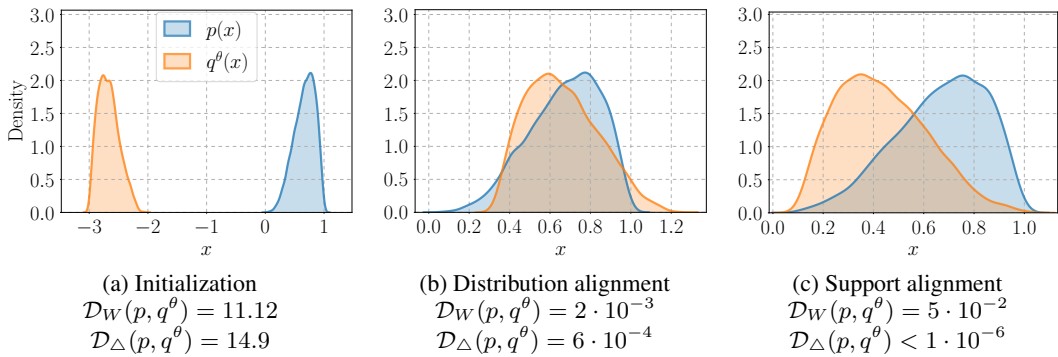

| (a) Initialization | (b) Distribution alignment | (c) Support alignment |
|---|---|---|
| $\mathcal{D}_W(p, q^\theta) = 11.12$ | $\mathcal{D}_W(p, q^\theta) = 2 \cdot 10^{-3}$ | $\mathcal{D}_W(p, q^\theta) = 5 \cdot 10^{-2}$ |
| $\mathcal{D}_\triangle(p, q^\theta) = 14.9$ | $\mathcal{D}_\triangle(p, q^\theta) = 6 \cdot 10^{-4}$ | $\mathcal{D}_\triangle(p, q^\theta) < 1 \cdot 10^{-6}$ |

Figure 1: Illustration of differences between the final configurations of distribution alignment and support alignment procedures. $p(x)$ is a fixed Beta distribution $p(x) = \text{Beta}(x \mid 4, 2)$ with support $[0, 1]$; $q^\theta(x)$ is a "shifted" Beta distribution $q^\theta(x) = \text{Beta}(x - \theta \mid 2, 4)$ parameterized by $\theta$ with support $[\theta, \theta + 1]$. Panel (a) shows the initial configuration with $\theta_{\text{init}} = -3$. Panel (b) shows the result by distribution alignment. Panel (c) shows the result by support alignment. We report Wasserstein distance $\mathcal{D}_W(p, q^\theta)$ (7) and SSD divergence $\mathcal{D}_\triangle(p, q^\theta)$ (1).

## 2 SSD DIVERGENCE AND SUPPORT ALIGNMENT

**Notation.** We consider an Euclidean space $\mathcal{X} = \mathbb{R}^n$ equipped with Borel sigma algebra $\mathcal{B}$ and a metric $d : \mathcal{X} \times \mathcal{X} \to \mathbb{R}$ (e.g. Euclidean distance). Let $\mathcal{P}$ be the set of probability measures on $(\mathcal{X}, \mathcal{B})$. For $p \in \mathcal{P}$, the support of $p$ is denoted by $\text{supp}(p)$ and is defined as the smallest closed set $X \subseteq \mathcal{X}$ such that $p(X) = 1$. $f_\sharp p$ denotes the pushforward measure of $p$ induced by a measurable mapping $f$. With a slight abuse of notation, we use $p(x)$ and $[f_\sharp p](t)$ to denote the densities of measures $p$ and $f_\sharp p$ evaluated at $x$ and $t$ respectively, implicitly assuming that the measures are absolutely continuous. The distance between a point $x \in \mathcal{X}$ and a subset $Y \subseteq \mathcal{X}$ is defined as $d(x, Y) = \inf_{y \in Y} d(x, y)$. The symmetric difference of two sets $A$ and $B$ is defined as $A \triangle B = (A \setminus B) \cup (B \setminus A)$.

### 2.1 DIFFERENCE BETWEEN SUPPORTS

To align the supports of distributions, we first need to evaluate how different they are. Similar to distribution divergences like Jensen–Shannon divergence, we introduce a notion of support divergence. A *support divergence*[2] between two distributions in $\mathcal{P}$ is a function $\mathcal{D}_S(\cdot, \cdot) : \mathcal{P} \times \mathcal{P} \to \mathbb{R}$ satisfying: 1) $\mathcal{D}_S(p, q) \geq 0$ for all $p, q \in \mathcal{P}$; 2) $\mathcal{D}_S(p, q) = 0$ iff $\text{supp}(p) = \text{supp}(q)$.

While a distribution divergence is sensitive to both density and support differences, a support divergence only needs to detect mismatches in supports, which are subsets of the metric space $\mathcal{X}$.

---

[2]It is not technically a divergence on the space of distributions, since $\mathcal{D}_S(p, q) = 0$ does not imply $p = q$.

An example of a distance between subsets of a metric space is the Hausdorff distance: $d_H(X,Y) = \max\{\sup_{x\in X} d(x,Y), \sup_{y\in Y} d(y,X)\}$. Since it depends only on the greatest distance between a point and a set, minimizing this objective for alignment only provides signal to a single point. To make the optimization less sparse, we consider all points that violate the support alignment criterion and introduce *symmetric support difference* (SSD) divergence:

$$\mathcal{D}_\triangle(p,q) = \mathbb{E}_{x\sim p}\left[d(x, \text{supp}(q))\right] + \mathbb{E}_{x\sim q}\left[d(x, \text{supp}(p))\right]. \tag{1}$$

**Proposition 2.1.** *SSD divergence $\mathcal{D}_\triangle(p,q)$ is a support divergence.*

We note that our proposed SSD divergence is closely related to Chamfer distance/divergence (CD) (Fan et al., 2017; Nguyen et al., 2021) and Relaxed Word Mover's Distance (RWMD) (Kusner et al., 2015). While both CD and RWMD are stated for discrete points (see Section 6 for further comments), SSD divergence is a general difference measure between arbitrary (discrete or continuous) distributions. This distinction, albeit small, is important in our theoretical analysis (Sections 2.2, 4.1).

## 2.2 Support alignment in one-dimensional space

Goodfellow et al. (2014) showed that the log-loss discriminator $f : \mathcal{X} \to [0,1]$, trained to distinguish samples from distributions $p$ and $q$ ($\sup_f \mathbb{E}_{x\sim p}[\log f(x)] + \mathbb{E}_{x\sim q}[\log(1-f(x))]$) can be used to estimate the Jensen–Shannon divergence between $p$ and $q$. The closed form maximizer $f^*$ is

$$f^*(x) = \frac{p(x)}{p(x)+q(x)}, \qquad \forall x \in \text{supp}(p) \cup \text{supp}(q). \tag{2}$$

Note that for a point $x \notin \text{supp}(p) \cup \text{supp}(q)$ the value of $f^*(x)$ can be set to an arbitrary value in $[0,1]$, since the log-loss does not depend on $f(x)$ for such $x$. The form of the optimal discriminator (2) gives rise to our main theorem below, which characterizes the ability of the log-loss discriminator to identify support misalignment.

**Theorem 2.1.** *Let $p$ and $q$ be the distributions with densities satisfying*

$$\frac{1}{C} < p(x) < C, \quad \forall\, x \in \text{supp}(p); \qquad \frac{1}{C} < q(x) < C, \quad \forall\, x \in \text{supp}(q). \tag{3}$$

*Let $f^*$ be the optimal discriminator (2). Then, $\mathcal{D}_\triangle(p,q) = 0$ if and only if $\mathcal{D}_\triangle(f^*_\sharp p, f^*_\sharp q) = 0$.*

The idea of the proof is to show that the extreme values (0 and 1) of $f^*(x)$ can only be attained in $x \in \text{supp}(p) \triangle \text{supp}(q)$. Assumption (3) guarantees that $f^*(x)$ cannot approach neither 0 nor 1 in the intersection of the supports $\text{supp}(p) \cap \text{supp}(q)$, i.e. the values $\{f^*(x) \,|\, x \in \text{supp}(p) \cap \text{supp}(q)\}$ are separated from the extreme values 0 and 1.

We conclude this section with two technical remarks on Theorem 2.1.

**Remark 2.1.1.** The result of Theorem 2.1 does not necessarily hold for other types of discriminators. For instance, the dual Wasserstein discriminator (Arjovsky et al., 2017; Gulrajani et al., 2017) does not always highlight the support difference in the original space as a support difference in the discriminator output space. This observation is formaly stated in the following proposition.

**Proposition 2.2.** *Let $f^\star_W$ be the maximizer of $\sup_{f:L(f)\le 1} \mathbb{E}_{x\sim p}[f(x)] - \mathbb{E}_{x\sim q}[f(x)]$, where $L(\cdot)$ is the Lipschitz constant. There exist $p$ and $q$ with $\text{supp}(p) \ne \text{supp}(q)$ but $\text{supp}(f^\star_{W\sharp} p) = \text{supp}(f^\star_{W\sharp} q)$.*

**Remark 2.1.2.** In practice the discriminator is typically parameterized as $f(x) = \sigma(g(x))$, where $g : \mathcal{X} \to \mathbb{R}$ is realized by a deep neural network and $\sigma(x) = (1+e^{-x})^{-1}$ is the sigmoid function. The optimization problem for $g$ is

$$\inf_g \mathbb{E}_{x\sim p}\left[\log(1+e^{-g(x)})\right] + \mathbb{E}_{x\sim q}\left[\log(1+e^{g(x)})\right], \tag{4}$$

and the optimal solution is $g^*(x) = \log p(x) - \log q(x)$. Naturally the result of Theorem 2.1 holds for $g^*$, since $g^*(x) = \sigma^{-1}(f^*(x))$ and $\sigma$ is a bijective mapping from $\mathbb{R} \cup \{-\infty, \infty\}$ to $[0,1]$.

## 3 ADVERSARIAL SUPPORT ALIGNMENT

We consider distributions $p$ and $q$ parameterized by $\theta$: $p^\theta, q^\theta$. The log-loss discriminator $g$ optimized for (4) is parameterized by $\psi$: $g^\psi$. Our analysis in Section 2.2 already suggests an algorithm. Namely, we can optimize $\theta$ by minimizing $\mathcal{D}_\triangle(g^\psi_\sharp p^\theta, g^\psi_\sharp q^\theta)$ while optimizing $\psi$ by (4). This adversarial game is analogous to the setup of the existing distribution alignment algorithms[3].

In practice, rather than having direct access to $p^\theta, q^\theta$, which is unavailable, we are often given i.i.d. samples $\{x_i^p\}_{i=1}^N, \{x_i^q\}_{i=1}^M$. They form discrete distributions $\hat{p}^\theta(x) = \frac{1}{N}\sum_{i=1}^N \delta(x - x_i^p)$, $\hat{q}^\theta(x) = \frac{1}{M}\sum_{i=1}^M \delta(x - x_i^q)$, and $[g^\psi_\sharp \hat{p}^\theta](t) = \frac{1}{N}\sum_{i=1}^N \delta(t - g^\psi(x_i^p))$, $[g^\psi_\sharp \hat{q}^\theta](t) = \frac{1}{M}\sum_{i=1}^M \delta(t - g^\psi(x_i^q))$. Since $g^\psi_\sharp \hat{p}^\theta$ and $g^\psi_\sharp \hat{q}^\theta$ are discrete distributions, they have supports $\{g^\psi(x_i^p)\}_{i=1}^N$ and $\{g^\psi(x_i^q)\}_{i=1}^M$ respectively. SSD divergence between discrete distributions $g^\psi_\sharp \hat{p}^\theta$ and $g^\psi_\sharp \hat{q}^\theta$ is

$$\mathcal{D}_\triangle(g^\psi_\sharp \hat{p}^\theta, g^\psi_\sharp \hat{q}^\theta) = \frac{1}{N}\sum_{i=1}^N d\left(g^\psi(x_i^p), \{g^\psi(x_j^q)\}_{j=1}^M\right) + \frac{1}{M}\sum_{i=1}^M d\left(g^\psi(x_i^q), \{g^\psi(x_j^p)\}_{j=1}^N\right). \quad (5)$$

**Effect of mini-batch training.** When training on large datasets, we need to rely on stochastic optimization with mini-batches. We denote the mini-batches (of same size, as in common practice) from $p^\theta$ and $q^\theta$ as $x^p = \{x_i^p\}_{i=1}^m$ and $x^q = \{x_i^q\}_{i=1}^m$ respectively. By minimizing $\mathcal{D}_\triangle(g^\psi(x^p), g^\psi(x^q))$, we only consider the mini-batch support distance rather than the population support distance (5). We observe that in practice the described algorithm brings the distributions to a state closer to distribution alignment rather than support alignment (see Appendix D.5 for details). The problem is in the typically small batch size. The algorithm actually tries to enforce support alignment for all possible pairs of mini-batches, which is a much stricter constraint than population support alignment.

To address the issue mentioned above, without working with a much larger batch size, we create two "history buffers": $h^p$, storing the previous 1D discriminator outputs of (at most) $n$ samples from $p^\theta$, and a similar buffer $h^q$ for $q^\theta$. Specifically, $h = \{g^{\psi_{\text{old},i}}(x_{\text{old},i})\}_{i=1}^n$ stores the values of the previous $n$ samples $x_{\text{old},i}$ mapped by their corresponding past "versions" of the discriminator $g^{\psi_{\text{old},i}}$. We minimize $\mathcal{D}_\triangle(v^p, v^q)$, where $v^p = \text{concat}(h^p, g^\psi(x^p))$, $v^q = \text{concat}(h^q, g^\psi(x^q))$:

$$\mathcal{D}_\triangle(v^p, v^q) = \frac{1}{n+m}\left(\sum_{i=1}^{n+m} d(v_i^p, v^q) + \sum_{j=1}^{n+m} d(v_j^q, v^p)\right). \quad (6)$$

Note that $\mathcal{D}_\triangle(\cdot, \cdot)$ between two sets of 1D samples can be efficiently calculated since $d(v_i^p, v^q)$ and $d(v_j^q, v^p)$ are simply 1-nearest neighbor distances in 1D. Moreover the history buffers store only the scalar values from the previous batches. These values are only considered in nearest neighbor assignment but do not directly provide gradient signal for optimization. Thus, the computation overhead of including a long history buffer is very light. We present our full algorithm, *Adversarial Support Alignment* (ASA), in Algorithm 1.

## 4 SPECTRUM OF NOTIONS OF ALIGNMENT

In this section, we take a closer look into our work and different existing notions of alignment that have been proposed in the literature, especially their formulations from the optimal transport perspective. We show that our proposed support alignment framework is a limit of existing notions of alignment, both in terms of theory and algorithm, by increasing transportation assignment tolerance.

### 4.1 THEORETICAL CONNECTIONS

**Distribution alignment.** Wasserstein distance is a commonly used objective for distribution alignment. In our analysis, we focus on the Wasserstein-1 distance:

$$\mathcal{D}_W(p, q) = \inf_{\gamma \in \Gamma(p,q)} \mathbb{E}_{(x,y)\sim\gamma}[d(x, y)], \quad (7)$$

---

[3]Following existing adversarial distribution alignment methods, e.g. (Goodfellow et al., 2014), we use single update of $\psi$ per 1 update of $\theta$. While theoretical analysis for both distribution alignment and support alignment (ours) assume optimal discriminators, training with single update of $\psi$ is computationally cheap and effective.

---

**Algorithm 1** Our proposed ASA algorithm. $n$ (maximum history buffer size), we use $n = 1000$.

1: **for** number of training steps **do**
2:     Sample mini-batches $\{x_i^p\}_{i=1}^m \sim p^\theta$, $\{x_i^q\}_{i=1}^m \sim q^\theta$.
3:     Perform optimization step on $\psi$ using stochastic gradient

$$\nabla_\psi \left( \frac{1}{m} \sum_{i=1}^m \left[ \log(1 + \exp(-g^\psi(x_i^p))) + \log(1 + \exp(g^\psi(x_i^q))) \right] \right).$$

4:     $v^p \leftarrow \text{concat}(h^p, \{g^\psi(x_i^p)\}_{i=1}^m)$, $v^q \leftarrow \text{concat}(h^q, \{g^\psi(x_i^q)\}_{i=1}^m)$.
5:     $\pi_{p\to q}^i \leftarrow \arg\min_j d(v_i^p, v_j^q)$, $\pi_{q\to p}^j \leftarrow \arg\min_i d(v_i^p, v_j^q)$.
6:     Perform optimization step on $\theta$ using stochastic gradient

$$\nabla_\theta \left( \frac{1}{n+m} \sum_{i=1}^{n+m} \left[ d(v_i^p, v_{\pi_{p\to q}^i}^q) + d(v_i^q, v_{\pi_{q\to p}^i}^p) \right] \right).$$

7:     UPDATEHISTORY$(h^p, \{g^\psi(x_i^p)\}_{i=1}^m)$, UPDATEHISTORY$(h^q, \{g^\psi(x_i^q)\}_{i=1}^m)$.
8: **end for**

---

where $\Gamma(p, q)$ is the set of all measures on $\mathcal{X} \times \mathcal{X}$ with marginals of $p$ and $q$, respectively. The value of $\mathcal{D}_W(p, q)$ is the minimal transportation cost for transporting probability mass from $p$ to $q$. The transportation cost is zero if and only if $p = q$, meaning the distributions are aligned.

**Relaxed distribution alignment.** Wu et al. (2019b) proposed a modified Wasserstein distance to achieve asymmetrically-relaxed distribution alignment, namely $\beta$-admissible Wasserstein distance:

$$\mathcal{D}_W^\beta(p, q) = \inf_{\gamma \in \Gamma_\beta(p,q)} \mathbb{E}_{(x,y)\sim\gamma}[d(x, y)], \tag{8}$$

where $\Gamma_\beta(p, q)$ is the set of all measures $\gamma$ on $\mathcal{X} \times \mathcal{X}$ such that $\int \gamma(x, y)dy = p(x), \forall x$ and $\int \gamma(x, y)dx \le (1 + \beta)q(y), \forall y$. With the relaxed marginal constraints, one could choose a transportation plan $\gamma$ which transports probability mass from $p$ to a modified distribution $q'$ rather than the original distribution $q$ as long as $q'$ satisfies the constraint $q'(x) \le (1 + \beta)q(x), \forall x$. Therefore, $\mathcal{D}_W^\beta(p, q)$ is zero if and only if $p(x) \le (1 + \beta)q(x), \forall x$. In (Wu et al., 2019b), $\beta$ is normally set to a positive finite number to achieve the asymmetric-relaxation of distribution alignment, and it is shown that $\mathcal{D}_W^0(p, q) = \mathcal{D}_W(p, q)$. We can extend $\mathcal{D}_W^\beta(p, q)$ to a symmetric version, which we term $\beta_1, \beta_2$-admissible Wasserstein distance:

$$\mathcal{D}_W^{\beta_1, \beta_2}(p, q) = \mathcal{D}_W^{\beta_1}(p, q) + \mathcal{D}_W^{\beta_2}(q, p). \tag{9}$$

The aforementioned property of $\beta$-admissible Wasserstein distance implies that $\mathcal{D}_W^{\beta_1, \beta_2}(p, q) = 0$ if and only if $p(x) \le (1 + \beta_1)q(x), \forall x$ and $q(x) \le (1 + \beta_2)p(x), \forall x$, in which case we call $p$ and $q$ "$(\beta_1, \beta_2)$-aligned", with $\beta_1$ and $\beta_2$ controlling the transportation assignment tolerances.

**Support alignment.** The term $\mathbb{E}_p[d(x, \text{supp}(q))]$ in (1) represents the average distance from samples in $p$ to the support of $q$. From the optimal transport perspective, this value is the minimal transportation cost of transporting the probability mass of $p$ into the support of $q$. We show that SSD divergence can be considered as a transportation cost in the limit of infinite assignment tolerance.

**Proposition 4.1.** $\mathcal{D}_W^{\infty, \infty}(p, q) := \lim_{\beta_1, \beta_2 \to \infty} \mathcal{D}_W^{\beta_1, \beta_2}(p, q) = \mathcal{D}_\triangle(p, q)$.

We now have completed the spectrum of alignment objectives defined within the optimal transport framework. The following proposition establishes the relationship within the spectrum.

**Proposition 4.2.** *Let $p$ and $q$ be two distributions in $\mathcal{P}$. Then,*

1. $\mathcal{D}_W(p, q) = 0$ *implies* $\mathcal{D}_W^{\beta_1, \beta_2}(p, q) = 0$ *for all finite* $\beta_1, \beta_2 > 0$.

2. $\mathcal{D}_W^{\beta_1, \beta_2}(p, q) = 0$ *for some finite* $\beta_1, \beta_2 > 0$ *implies* $\mathcal{D}_\triangle(p, q) = 0$.

3. *The converse of statements 1 and 2 are false.*

In addition to the result presented in Theorem 2.1, we can show that the log-loss discriminator can also "preserve" the existing notions of alignment.

**Proposition 4.3.** *Let $f^*$ be the optimal discriminator (2) for given distributions $p$ and $q$. Then,*

1. $\mathcal{D}_W(p, q) = 0$ *iff* $\mathcal{D}_W(f^*{}_\sharp p, f^*{}_\sharp q) = 0$;     2. $\mathcal{D}_W^{\beta_1, \beta_2}(p, q) = 0$ *iff* $\mathcal{D}_W^{\beta_1, \beta_2}(f^*{}_\sharp p, f^*{}_\sharp q) = 0$.

## 4.2 Algorithmic connections

The result of Proposition 4.3 suggests methods similar to our ASA algorithm presented in Section 3 can achieve different notions of alignment by minimizing objectives discussed in Section 4.1 between the 1D pushforward distributions. We consider the setup used in Section 3 but without history buffers to simplify the analysis, as their usage is orthogonal to our discussion in this section.

Recall that we work with a mini-batch setting, where $\{x_i^p\}_{i=1}^m$ and $\{x_i^q\}_{i=1}^m$ are sampled from $p$ and $q$ respectively, and $g$ is the adversarial log-loss discriminator. We denote the corresponding 1D outputs from the log-loss discriminator by $o^p = \{o_i^p\}_{i=1}^m = \{g(x_i^p)\}_{i=1}^m$ and $o^q$ (defined similarly).

**Distribution alignment.** We adapt (7) for $\{o_i^p\}_{i=1}^m$ and $\{o_i^q\}_{i=1}^m$:

$$\mathcal{D}_W(o^p, o^q) = \inf_{\gamma \in \Gamma(o^p, o^q)} \frac{1}{m} \sum_{i=1}^m \sum_{j=1}^m \gamma_{ij} d(o_i^p, o_j^q), \qquad (10)$$

where $\Gamma(o^p, o^q)$ is the set of $m \times m$ doubly stochastic matrices. Since $o^p$ and $o^q$ are sets of 1D samples with the same size, it can be shown (Rabin et al., 2011) that the optimal $\gamma^*$ corresponds to an assignment $\pi^*$, which pairs points in the sorting order and can be computed efficiently by sorting both sets $o^p$ and $o^q$. The transportation cost is zero if and only if there exists an invertible 1-to-1 assignment $\pi^*$ such that $o_i^p = o_{\pi^*(i)}^q$. GAN training algorithms proposed in (Deshpande et al., 2018; 2019) utilize the above sorting procedure to estimate the maximum sliced Wasserstein distance.

**Relaxed distribution alignment.** Similarly, we can adapt (8):

$$\mathcal{D}_W^\beta(o^p, o^q) = \inf_{\gamma \in \Gamma_\beta(o^p, o^q)} \frac{1}{m} \sum_{i=1}^m \sum_{j=1}^m \gamma_{ij} d(o_i^p, o_j^q), \qquad (11)$$

where $\Gamma_\beta(o^p, o^q)$ is the set of $m \times m$ matrices with non-negative real entries, such that $\sum_{j=1}^m \gamma_{ij} = 1, \forall i$ and $\sum_{i=1}^m \gamma_{ij} \leq 1 + \beta, \forall j$. The optimization goal in (11) is to find a "soft-assignment" $\gamma$ which describes the transportation of probability mass from points $o_i^p$ in $o^p$ to points $o_i^q$ in $o^q$. The parameter $\beta$ controls the set of admissible assignments $\Gamma_\beta$, which is similar to its role discussed in Section 4.1: with transportation assignment tolerance $\beta$, the total mass of points in $o^p$ transported to each of the points $o_i^q$ cannot exceed $1 + \beta$. We refer to such assignments as $(\beta + 1)$-to-1 assignment. The transportation cost is zero if and only if there exists such an assignment between $o^p$ and $o^q$.

It can be shown (see Appendix C) that for integer value of $\beta$, the set of minimizers of (11) must contain a "hard-assignment" transportation plan, which assigns each point $o_i^p$ to exactly one point $o_j^q$. Then $(1 + \beta)$ gives the upper bound on the number of points $o_i^p$ that can be transported to given point $o_j^q$. This hard assignment problem can be solved quasi-linearly with worst case time complexity $\mathcal{O}\left((\beta + 1)m^2\right)$ (Bonneel & Coeurjolly, 2019), which, combined with Proposition 4.3, can lead to new algorithms for relaxed distribution alignment besides those proposed in Wu et al. (2019b).

**Support alignment.** When $\beta = \infty$, the sum $\sum_{i=1}^m \gamma_{ij}$ is unconstrained for all $j$, and each point $o_i^p$ can be assigned to any of the points $o_j^q$. The optimal solution is simply 1-nearest neighbor assignment, or to follow the above terminology, $\infty$-to-1 assignment.

## 5 Experiments

**Problem setting.** We evaluate our proposed ASA method in the setting of unsupervised domain adaptation (UDA). The goal of UDA algorithms is to train and "adapt" a classification model $M : \mathcal{X} \to \mathcal{Y}$ from source domain distribution $p_{X,Y}$ to target domain distribution $q_{X,Y}$ given the access to a labeled source dataset $\{x_i^p, y_i^p\}_{i=1}^{N^p} \sim p_{X,Y}$ and an unlabeled target dataset $\{x_i^q\}_{i=1}^{N^q} \sim q_X$.

A common approach for UDA is to represent $M$ as $C^\phi \circ F^\theta$: a classifier $C^\phi : \mathcal{Z} \to \mathcal{Y}$ and a feature extractor $F^\theta : \mathcal{X} \to \mathcal{Z}$, and train $C^\phi$ and $F^\theta$ by minimizing: 1) classification loss $\ell_{\text{cls}}$ on source examples; 2) alignment loss $\mathcal{D}_{\text{align}}$ measuring discrepancy between $p_Z^\theta = F^\theta_\sharp p_X$ and $q_Z^\theta = F^\theta_\sharp q_X$:

$$\min_{\phi, \theta} \ \frac{1}{N^p} \sum_{i=1}^{N^p} \ell_{\text{cls}}(C^\phi(F^\theta(x_i^p)), y_i^p) + \lambda \cdot \mathcal{D}_{\text{align}}\left(\{F^\theta(x_i^p)\}_{i=1}^{N^p}, \{F^\theta(x_i^q)\}_{i=1}^{N^q}\right), \qquad (12)$$

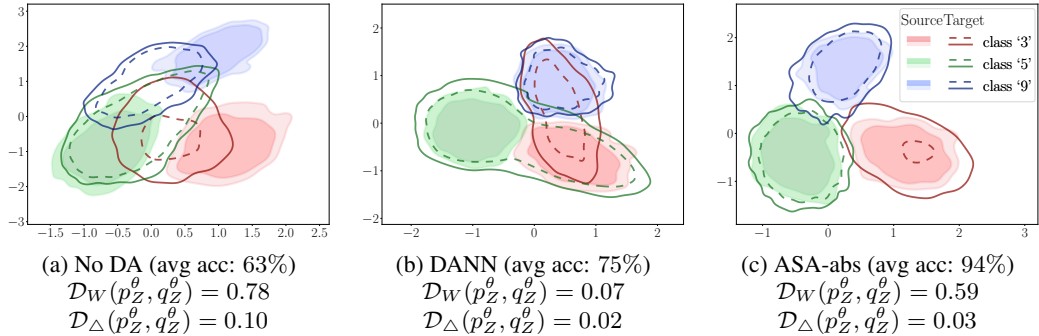

(a) No DA (avg acc: 63%)
$\mathcal{D}_W(p_Z^\theta, q_Z^\theta) = 0.78$
$\mathcal{D}_\triangle(p_Z^\theta, q_Z^\theta) = 0.10$

(b) DANN (avg acc: 75%)
$\mathcal{D}_W(p_Z^\theta, q_Z^\theta) = 0.07$
$\mathcal{D}_\triangle(p_Z^\theta, q_Z^\theta) = 0.02$

(c) ASA-abs (avg acc: 94%)
$\mathcal{D}_W(p_Z^\theta, q_Z^\theta) = 0.59$
$\mathcal{D}_\triangle(p_Z^\theta, q_Z^\theta) = 0.03$

Figure 2: Visualization of learned 2D embeddings on 3-class USPS→MNIST with label distribution shift. In source domain, all classes have equal probability $\frac{1}{3}$. The target probabilities of classes '3', '5', '9' are $[23\%, 65\%, 12\%]$. Each panel shows 2 level sets (outer one approximates the support) of the kernel density estimates of embeddings in source (filled regions) and target domains (solid/dashed lines). We report the average class accuracy of the target domain, $\mathcal{D}_W$ and $\mathcal{D}_\triangle$ between embeddings.

In practice $\mathcal{D}_{\text{align}}$ is an estimate of a divergence measure via an adversarial discriminator $g^\psi$. Choices of $\mathcal{D}_{\text{align}}$ include f-divergences (Ganin et al., 2016; Nowozin et al., 2016) and Wasserstein distance (Arjovsky et al., 2017) to enforce distribution alignment and versions of re-weighted/relaxed distribution divergences (Wu et al., 2019b; Tachet des Combes et al., 2020) to enforce relaxed distribution alignment. For support alignment, we apply the proposed ASA method as the alignment subroutine in (12) with log-loss discriminator $g^\psi$ (4) and $\mathcal{D}_{\text{align}}$ computed as (6).

**Task specifications.** We consider 3 UDA tasks: USPS→MNIST, STL→CIFAR, and VisDA-2017, and 2 versions of ASA: **ASA-sq**, **ASA-abs** corresponding to squared and absolute distances respectively for $d(\cdot, \cdot)$ in (6). We compare ASA with: **No DA** (no domain adaptation), **DANN** (Ganin et al., 2016) (distribution alignment with JS divergence), **VADA** (Shu et al., 2018) (distribution alignment with virtual adversarial training), **IWDAN**, **IWCDAN** (Tachet des Combes et al., 2020) (relaxed distribution alignment via importance weighting) **sDANN-$\beta$** (Wu et al., 2019b) (relaxed/$\beta$-admissible JS divergence via re-weighting). Please refer to Appendix D for full experimental details.

To evaluate the robustness of the methods, we simulate label distribution shift by subsampling source and target dataset, so that source has balanced label distribution and target label distribution follows the power law $q_Y(y) \propto \sigma(y)^{-\alpha}$, where $\sigma$ is a random permutation of class labels $\{1, \ldots, K\}$ and $\alpha$ controls the severity of the shift ($\alpha = 0$ means balanced label distribution). For each task, we generate 5 random permutations $\sigma$ for 4 different shift levels $\alpha \in \{0, 1, 1.5, 2\}$. Essentially we transform each (source, target) dataset pair to $5 \times 4 = 20$ tasks of different difficulty levels, since classes are not equally difficult and different permutations can give them different weights.

**Evaluation metrics.** We choose the average (per-)class accuracy and minimum (per-)class accuracy on the target test set as evaluation metrics. Under the average class accuracy metric, all classes are treated as equally important (despite the unequal representation during training for $\alpha > 0$), and the minimum class accuracy focuses on model's worst within-class performance. In order to account for the variability of task difficulties across random permutations of target labels, we report robust statistics, median and a 25-75 percentile interval, across 5 runs.

**Illustrative example.** First we consider a simplified setting to intuitively understand and directly analyze the behavior of our proposed support alignment method in domain adaptation under label distribution shift. We consider a 3-class USPS→MNIST problem by selecting a subset of examples corresponding to digits '3', '5', and '9', and use a feature extractor network with 2D output space. We introduce label distribution shift as described above with $\alpha = 1.5$, i.e. the probabilities of classes in the target domain are $12\%$, $23\%$, and $65\%$. We compare No DA, DANN, ASA-abs by their average target classification accuracy, Wasserstein distance $\mathcal{D}_W(p_Z^\theta, q_Z^\theta)$ and SSD divergence $\mathcal{D}_\triangle(p_Z^\theta, q_Z^\theta)$ between the learned embeddings of source and target domain. We apply a global affine transformation to each embedding space in order to have comparable distances between different spaces: we center the embeddings so that their average is 0 and re-scale them so that their average norm is 1. The

Table 1: Average and minimum class accuracy (%) on USPS→MNIST with different levels of shifts in label distributions (higher $\alpha$ implies more severe imbalance). We report median (the main number), and 25 (subscript) and 75 (superscript) percentiles across 5 runs.

| Algorithm | $\alpha = 0.0$ | | $\alpha = 1.0$ | | $\alpha = 1.5$ | | $\alpha = 2.0$ | |
|---|---|---|---|---|---|---|---|---|
| | average | min | average | min | average | min | average | min |
| No DA | $71.9_{70.4}^{72.9}$ | $20.3_{17.6}^{22.9}$ | $72.9_{72.0}^{74.7}$ | $25.8_{18.3}^{31.8}$ | $71.3_{71.2}^{72.5}$ | $27.5_{24.2}^{37.3}$ | $71.3_{70.6}^{73.0}$ | $16.6_{10.8}^{26.8}$ |
| DANN | $97.8_{97.6}^{97.8}$ | $96.0_{95.8}^{96.1}$ | $83.5_{76.7}^{84.6}$ | $25.1_{08.4}^{36.9}$ | $70.0_{63.9}^{71.2}$ | $01.1_{01.0}^{01.5}$ | $57.8_{52.0}^{60.4}$ | $00.9_{00.5}^{01.6}$ |
| VADA | $\mathbf{98.0}_{97.9}^{98.0}$ | $96.2_{95.9}^{96.3}$ | $88.2_{88.1}^{89.9}$ | $48.9_{47.8}^{50.0}$ | $78.2_{70.7}^{83.1}$ | $06.6_{02.4}^{23.5}$ | $61.9_{56.3}^{65.4}$ | $01.4_{00.8}^{01.5}$ |
| IWDAN | $97.5_{97.4}^{97.5}$ | $95.7_{95.7}^{95.9}$ | $95.7_{92.6}^{95.8}$ | $81.3_{67.1}^{82.3}$ | $86.5_{80.2}^{87.8}$ | $15.2_{04.2}^{55.0}$ | $74.4_{70.0}^{78.6}$ | $07.3_{06.3}^{22.4}$ |
| IWCDAN | $\mathbf{98.0}_{97.9}^{98.1}$ | $\mathbf{96.6}_{96.4}^{96.9}$ | $\mathbf{96.7}_{93.3}^{97.5}$ | $85.1_{65.3}^{93.9}$ | $91.3_{90.5}^{93.8}$ | $66.5_{64.1}^{74.5}$ | $77.5_{77.3}^{82.3}$ | $22.2_{02.7}^{45.4}$ |
| sDANN-4 | $87.4_{87.2}^{95.7}$ | $05.6_{05.6}^{90.0}$ | $94.9_{94.7}^{94.9}$ | $\mathbf{85.7}_{84.4}^{87.7}$ | $86.8_{85.5}^{89.1}$ | $21.6_{15.4}^{50.3}$ | $81.5_{81.3}^{83.1}$ | $39.3_{37.9}^{56.2}$ |
| ASA-sq | $93.7_{93.3}^{93.9}$ | $89.2_{88.4}^{89.4}$ | $92.3_{91.5}^{93.6}$ | $83.5_{80.8}^{88.7}$ | $90.9_{89.6}^{92.1}$ | $69.9_{66.6}^{82.0}$ | $87.2_{85.8}^{89.3}$ | $62.5_{46.4}^{69.3}$ |
| ASA-abs | $94.1_{93.8}^{94.5}$ | $88.9_{87.0}^{91.2}$ | $92.8_{89.3}^{93.2}$ | $78.9_{65.1}^{82.9}$ | $\mathbf{92.5}_{90.9}^{92.9}$ | $\mathbf{82.4}_{74.5}^{85.4}$ | $\mathbf{90.4}_{89.2}^{90.7}$ | $\mathbf{68.4}_{67.5}^{73.0}$ |

Table 2: Results on STL→CIFAR. Same setup and reporting metrics as Table 1.

| Algorithm | $\alpha = 0.0$ | | $\alpha = 1.0$ | | $\alpha = 1.5$ | | $\alpha = 2.0$ | |
|---|---|---|---|---|---|---|---|---|
| | average | min | average | min | average | min | average | min |
| No DA | $69.9_{69.8}^{70.0}$ | $49.8_{45.3}^{50.6}$ | $68.8_{68.3}^{69.3}$ | $47.2_{45.3}^{48.2}$ | $66.8_{66.4}^{67.2}$ | $46.0_{45.8}^{47.0}$ | $65.8_{64.8}^{66.7}$ | $43.7_{41.6}^{44.6}$ |
| DANN | $75.3_{74.9}^{75.4}$ | $54.6_{54.2}^{56.6}$ | $69.9_{68.6}^{70.1}$ | $44.8_{40.7}^{45.1}$ | $64.9_{63.7}^{67.1}$ | $34.9_{33.9}^{36.8}$ | $63.3_{57.4}^{64.8}$ | $27.0_{21.2}^{28.5}$ |
| VADA | $\mathbf{76.7}_{76.6}^{76.7}$ | $\mathbf{56.9}_{53.5}^{58.3}$ | $70.6_{70.0}^{71.0}$ | $47.7_{44.0}^{48.8}$ | $66.1_{65.4}^{66.5}$ | $35.7_{33.3}^{39.3}$ | $63.2_{60.2}^{64.7}$ | $25.5_{25.2}^{28.0}$ |
| IWDAN | $69.9_{69.9}^{70.7}$ | $50.5_{47.9}^{50.6}$ | $68.7_{68.6}^{69.1}$ | $45.8_{44.8}^{50.5}$ | $67.1_{65.9}^{67.3}$ | $44.7_{40.4}^{44.8}$ | $64.4_{63.6}^{64.9}$ | $36.8_{34.5}^{37.9}$ |
| IWCDAN | $70.1_{70.1}^{70.2}$ | $47.8_{42.4}^{49.3}$ | $69.4_{69.1}^{69.4}$ | $47.1_{46.3}^{51.3}$ | $66.1_{65.0}^{67.2}$ | $39.9_{37.7}^{40.8}$ | $64.5_{63.9}^{65.1}$ | $37.0_{35.5}^{40.2}$ |
| sDANN-4 | $71.8_{71.7}^{72.1}$ | $52.1_{52.1}^{52.8}$ | $\mathbf{71.1}_{70.4}^{71.7}$ | $49.9_{48.1}^{51.8}$ | $69.4_{68.7}^{70.0}$ | $\mathbf{48.6}_{43.5}^{49.0}$ | $66.4_{66.2}^{67.9}$ | $39.0_{33.6}^{47.1}$ |
| ASA-sq | $71.7_{71.7}^{71.9}$ | $52.9_{46.7}^{53.4}$ | $70.7_{70.4}^{71.0}$ | $\mathbf{51.6}_{46.8}^{52.7}$ | $69.2_{69.2}^{69.3}$ | $45.6_{43.3}^{52.0}$ | $\mathbf{68.1}_{67.2}^{68.2}$ | $\mathbf{44.7}_{39.8}^{45.9}$ |
| ASA-abs | $71.6_{71.2}^{71.7}$ | $49.0_{48.4}^{53.5}$ | $70.9_{70.8}^{71.0}$ | $49.2_{47.3}^{50.0}$ | $\mathbf{69.6}_{69.6}^{69.9}$ | $43.2_{42.1}^{49.5}$ | $67.8_{66.6}^{68.2}$ | $40.9_{35.4}^{49.0}$ |

results are shown in Figure 2 and Table D.5. Compared to No DA, both DANN and ASA achieve support alignment. DANN enforces distribution alignment, and thus places some target embeddings into regions corresponding to the wrong class. In comparison, ASA does not enforce distribution alignment and maintains good class correspondence across the source and target embeddings.

**Main results.** The results of the main experimental evaluations are shown in Tables 1, 2, 3. Without any alignment, source only training struggles relatively to adapt to the target domain. Nonetheless, its performance across the imbalance levels remains robust, since the training procedure is the same. Agreeing with the observation and theoretical results from previous work (Zhao et al., 2019; Li et al., 2020; Tan et al., 2020; Wu et al., 2019b; Tachet des Combes et al., 2020), distribution alignment methods (DANN and VADA) perform well when there is no shift but suffer otherwise, whereas relaxed distribution alignment methods (IWDAN, IWCDAN and sDANN-$\beta$) show more resilience to shifts. On all tasks with positive $\alpha$, we observe that it is common for the existing methods to achieve good class average accuracies while suffering significantly on some individual classes. These results suggest that the often-ignored but important min-accuracy metric can be very challenging. Finally, our support alignment methods (ASA-sq and ASA-abs) are the most robust ones against the shifts, while still being competitive in the more balanced settings ($\alpha = 0$ or $1$). We achieve best results in the more imbalanced and difficult tasks ($\alpha = 1.5$ or $2$) for almost all categories on all datasets. Please refer to Appendix D for ablation studies and additional comparisons.

## 6  RELATED WORK

**Distribution alignment.** Apart from the works, e.g. (Ajakan et al., 2014; Ganin et al., 2016; Ganin & Lempitsky, 2015; Pei et al., 2018; Zhao et al., 2018; Long et al., 2018; Tachet des Combes et al.,

Table 3: Results on VisDA17. Same setup and reporting metrics as Table 1.

| Algorithm | $\alpha = 0.0$ average | min | $\alpha = 1.0$ average | min | $\alpha = 1.5$ average | min | $\alpha = 2.0$ average | min |
|---|---|---|---|---|---|---|---|---|
| No DA | $49.5\,^{50.5}_{49.4}$ | $22.2\,^{24.6}_{22.2}$ | $50.2\,^{50.8}_{49.2}$ | $21.2\,^{21.3}_{20.7}$ | $47.1\,^{47.6}_{46.6}$ | $18.6\,^{22.2}_{18.6}$ | $45.3\,^{46.5}_{45.2}$ | $19.5\,^{19.8}_{14.4}$ |
| DANN | $\mathbf{75.4}\,^{76.2}_{74.4}$ | $36.7\,^{40.9}_{35.6}$ | $64.1\,^{65.3}_{62.8}$ | $25.0\,^{29.3}_{24.8}$ | $52.1\,^{52.3}_{51.4}$ | $11.5\,^{12.4}_{11.4}$ | $43.1\,^{44.3}_{39.1}$ | $03.6\,^{14.3}_{03.6}$ |
| VADA | $75.3\,^{76.0}_{74.8}$ | $40.5\,^{41.8}_{39.7}$ | $64.6\,^{65.1}_{61.2}$ | $22.8\,^{28.2}_{21.7}$ | $53.0\,^{54.2}_{51.6}$ | $14.8\,^{21.7}_{13.7}$ | $43.9\,^{44.7}_{40.9}$ | $08.5\,^{11.1}_{05.0}$ |
| IWDAN | $73.2\,^{73.3}_{72.9}$ | $31.7\,^{34.8}_{22.8}$ | $64.4\,^{64.6}_{61.1}$ | $12.1\,^{24.7}_{05.0}$ | $51.3\,^{56.6}_{51.0}$ | $04.6\,^{10.4}_{02.1}$ | $45.1\,^{48.0}_{41.7}$ | $04.6\,^{13.6}_{01.2}$ |
| IWCDAN | $71.6\,^{75.2}_{70.6}$ | $27.6\,^{28.0}_{22.8}$ | $60.6\,^{61.0}_{60.2}$ | $01.1\,^{11.3}_{00.7}$ | $49.7\,^{51.9}_{45.6}$ | $02.2\,^{05.7}_{00.2}$ | $38.3\,^{46.2}_{37.3}$ | $00.6\,^{01.7}_{00.3}$ |
| sDANN-4 | $72.4\,^{73.3}_{71.8}$ | $37.8\,^{40.8}_{32.3}$ | $\mathbf{68.4}\,^{68.7}_{66.2}$ | $26.6\,^{29.4}_{26.2}$ | $57.2\,^{57.8}_{56.8}$ | $18.6\,^{23.9}_{16.7}$ | $50.7\,^{51.7}_{49.8}$ | $18.6\,^{20.0}_{17.1}$ |
| ASA-sq | $64.9\,^{65.0}_{63.7}$ | $35.7\,^{35.8}_{32.1}$ | $61.8\,^{63.2}_{60.6}$ | $\mathbf{31.4}\,^{34.4}_{20.4}$ | $\mathbf{57.8}\,^{58.3}_{55.5}$ | $\mathbf{26.7}\,^{32.1}_{17.3}$ | $51.9\,^{52.0}_{50.8}$ | $18.3\,^{21.2}_{16.9}$ |
| ASA-abs | $64.8\,^{65.0}_{64.5}$ | $\mathbf{40.6}\,^{41.9}_{36.0}$ | $62.0\,^{62.3}_{60.5}$ | $27.3\,^{29.7}_{16.7}$ | $57.1\,^{58.4}_{56.2}$ | $26.0\,^{31.2}_{13.9}$ | $\mathbf{52.5}\,^{56.6}_{51.9}$ | $\mathbf{19.7}\,^{22.2}_{17.7}$ |

2020; Li et al., 2018b; Tzeng et al., 2017; Shen et al., 2018; Kumar et al., 2018; Li et al., 2018a; Wang et al., 2021; Goodfellow et al., 2014; Arjovsky et al., 2017; Gulrajani et al., 2017; Mao et al., 2017; Radford et al., 2015; Salimans et al., 2018; Genevay et al., 2018; Wu et al., 2019a; Deshpande et al., 2018; 2019), that do distribution alignment, there are also papers (Long et al., 2015; 2017; Peng et al., 2019; Sun et al., 2016; Sun & Saenko, 2016) focusing on aligning some characteristics of the distribution, such as first or second moments. Our work is concerned with a different problem, support alignment, which is a novel objective in this line of work. In terms of methodology, our use of the discriminator output space to work with easier optimization in 1D is inspired by a line of work (Salimans et al., 2018; Genevay et al., 2018; Wu et al., 2019a; Deshpande et al., 2018; 2019) on sliced Wasserstein distance based models. Our result in Proposition 4.3 also provides theoretical insight on the practical effectiveness of 1D OT in (Deshpande et al., 2019).

**Relaxed distribution alignment.** In Section 4, we have already covered in detail the connections between our work and (Wu et al., 2019b). Balaji et al. (2020) introduced relaxed distribution alignment with a different focus, aiming to be insensitive to outliers. Chamfer distance/divergence (CD) is used to compute similarity between images/3D point clouds (Fan et al., 2017; Nguyen et al., 2021). For text data, Kusner et al. (2015) presented Relaxed Word Mover's Distance (RWMD) to prune candidates of similar documents. CD and RWMD are essentially the same as (5) with $d(\cdot, \cdot)$ being the Euclidean distance. They are computed by finding the nearest neighbor assignments. Our subroutine of calculating the support distance in the 1D discriminator output space is done similarly by finding nearest neighbors within the current batch and history buffers.

**Support estimation.** There exists a series of work, e.g. (Schölkopf et al., 2001; Hoffmann, 2007; Tax & Duin, 2004; Knorr et al., 2000; Chalapathy et al., 2017; Ruff et al., 2018; Perera et al., 2019; Deecke et al., 2018; Zenati et al., 2018), on novelty/anomaly detection problem, which can be casted as support estimation. We consider a fundamentally different problem setting. Our goal is to align the supports and our approach does not directly estimate the supports. Instead, we implicitly learn the relationships between supports (density ratio to be specific) via a discriminator.

## 7 CONCLUSION AND FUTURE WORK

In this paper, we studied the problem of aligning the supports of distributions. We formalized its theoretical connections with existing alignment notions and demonstrated the effectiveness of the approach in domain adaptation. We believe that our methodology opens possibilities for the design of more nuanced and structured alignment constraints, suitable for various use cases. One natural extension is support containment, achievable with only one term in (1). This approach is fitting for partial domain adaptation, where some source domain classes do not appear in the target domain. Another interesting direction is unsupervised domain transfer, where support alignment is more desired than existing distribution alignment methods due to mode imbalance (Binkowski et al., 2019).

ACKNOWLEDGMENTS

The computational experiments presented in this paper were performed on "Satori" cluster developed as a collaboration between MIT and IBM. We used Weights & Biases (Biewald, 2020) for experiment tracking and visualizations to develop insights for this paper.

TJ acknowledges support from MIT-IBM Watson AI Lab, from Singapore DSO, and MIT-DSTA Singapore collaboration. We thank Xiang Fu and all anonymous reviewers for their helpful comments regarding the paper's writing and presentation.

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

## A    PROOFS OF THE THEORETICAL RESULTS

### A.1    PROOF OF PROPOSITION 2.1

1) $\mathcal{D}_{\triangle}(p, q) \geq 0$ for all $p, q \in \mathcal{P}$:

Since $d(\cdot, \cdot) \geq 0$, for all $p, q$,

$$\mathrm{SD}(p, q) := \mathbb{E}_{x \sim p}[d(x, \mathrm{supp}(q))] = \mathbb{E}_{x \sim p}\left[\inf_{y \sim \mathrm{supp}(q)} d(x, y)\right] \geq 0, \tag{13}$$

which makes $\mathcal{D}_{\triangle}(p, q) = \mathrm{SD}(p, q) + \mathrm{SD}(q, p) \geq 0$.

---

2) $\mathcal{D}_{\triangle}(p, q) = 0$ if and only if $\mathrm{supp}(p) = \mathrm{supp}(q)$:

With statement 1, $\mathcal{D}_{\triangle}(p, q) = 0$ if and only if $\mathrm{SD}(p, q) = 0$ and $\mathrm{SD}(q, p) = 0$.

Then,

$$\mathrm{SD}(p, q) = 0 \implies \mathbb{E}_{x \sim p}[d(x, \mathrm{supp}(q))] = 0 \implies p\left(\{x | d(x, \mathrm{supp}(q)) > 0\}\right) = 0.$$

This is equivalent to
$$\forall x \in \mathrm{supp}(p), \ d(x, \mathrm{supp}(q)) = 0.$$

Thus, $\mathrm{supp}(p) \subseteq \mathrm{supp}(q)$, and similarly, $\mathrm{supp}(q) \subseteq \mathrm{supp}(p)$, which makes $\mathrm{supp}(p) = \mathrm{supp}(q)$.

### A.2    ASSUMPTION AND PROOF OF THEOREM 2.1

#### A.2.1    COMMENTS ON ASSUMPTION (3)

Assumption (3) is not restrictive. Indeed, distributions satisfying Assumption (3) include:

- uniform $p(x) = U(x; [a, b])$;
- truncated normal;
- $p(x)$ of the form
$$p(x) = \begin{cases} \frac{1}{Z_p} e^{-E_p(x)}, & x \in \mathrm{supp}(p), \\ 0, & x \notin \mathrm{supp}(p), \end{cases}$$
  with non-negative energy (unnormalized log-density) function $E_p : \mathcal{X} \to [0, \infty)$;
- mixture of any distributions satisfying Assumption (3), for instance the distributions shown in Figure A.1 top-left are mixtures of truncated normal distributions on $[-2, 2]$.

Starting from arbitrary density $p_0(x)$ with bounded support we can derive a density $p(x)$ satisfying Assumption (3) via density clipping and re-normalization

$$p(x) \propto \mathrm{clip}\left(p_0(x), \left[\frac{1}{C'}, C'\right]\right),$$

for some $C' > 1$.

#### A.2.2    PROOF OF THEOREM 2.1

First, we show that $\mathcal{D}_{\triangle}(p, q) = 0$ implies $\mathcal{D}_{\triangle}(f^*_{\sharp}p, f^*_{\sharp}q) = 0$.

$\mathcal{D}_{\triangle}(p, q) = 0$ implies $\mathrm{supp}(p) = \mathrm{supp}(q)$. Then for any mapping $f : \mathcal{X} \to \mathbb{R}$, we have $\mathrm{supp}(f_{\sharp}p) = \mathrm{supp}(f_{\sharp}q)$, which implies $\mathrm{supp}(f^*_{\sharp}p) = \mathrm{supp}(f^*_{\sharp}q)$. Thus, $\mathcal{D}_{\triangle}(f^*_{\sharp}p, f^*_{\sharp}q) = 0$.

---

Now, we prove that $\mathcal{D}_{\triangle}(f^*_{\sharp}p, f^*_{\sharp}q) = 0$ implies $\mathcal{D}_{\triangle}(p, q) = 0$ by contradiction.

$\mathcal{D}_\triangle(f^*{}_\sharp p, f^*{}_\sharp q) = 0$ implies the following:

$$\mathbb{E}_{t \sim f^*{}_\sharp p}\big[d(t, \operatorname{supp}(f^*{}_\sharp q)\big] = 0, \qquad \mathbb{E}_{t \sim f^*{}_\sharp q}\big[d(t, \operatorname{supp}(f^*{}_\sharp p)\big] = 0.$$

1) Suppose $\mathbb{E}_{x \sim p}[d(x, \operatorname{supp}(q))] > 0$. This is only possible if $p(\{x \mid x \in \operatorname{supp}(p) \setminus \operatorname{supp}(q)\}) > 0$. Since $x \in \operatorname{supp}(p) \setminus \operatorname{supp}(q)$ implies $p(x) > 0, q(x) = 0$, and for any $x \in \operatorname{supp}(p) \cup \operatorname{supp}(q)$, $p(x) > 0, q(x) = 0$ if and only if $f^*(x) = \frac{p(x)}{p(x)+q(x)} = 1$, we have:

$$\mathbb{P}_{f^*{}_\sharp p}(\{1\}) = \mathbb{P}_p(\{x \mid x \in \operatorname{supp}(p) \setminus \operatorname{supp}(q)\}) > 0,$$

and therefore $1 \in \operatorname{supp}(f^*{}_\sharp p)$.

For a real number $\alpha : 0 < \alpha < \frac{1}{C^2+1}$, consider the probability of the event $(1 - \alpha, 1] \subset [0, 1]$ under distribution $f^*{}_\sharp q$:

$$\mathbb{P}_{f^*{}_\sharp q}((1 - \alpha, 1]) = \mathbb{P}_q(\{x \mid f^*(x) \in (1 - \alpha, 1]\}).$$

By assumption (3), $p(x) < C$ and $q(x) > 0$ implies $q(x) > \frac{1}{C}$, therefore for $x : q(x) > 0$ we have

$$f^*(x) = \frac{p(x)}{p(x)+q(x)} < \frac{p(x)}{p(x)+\frac{1}{C}} < \frac{C}{C+\frac{1}{C}} = 1 - \frac{1}{C^2+1} < 1 - \alpha.$$

This means that $\mathbb{P}_{f^*{}_\sharp q}((1 - \alpha, 1]) = 0$, i.e. $\operatorname{supp}(f^*{}_\sharp q) \cap (1 - \alpha, 1] = \varnothing$.

To summarize, starting from the assumption that $\mathbb{E}_{x \sim p}[d(x, \operatorname{supp}(q))] > 0$ we showed that

- $1 \in \operatorname{supp}(f^*{}_\sharp p), \mathbb{P}_{f^*{}_\sharp p}(\{1\}) > 0$;
- $\operatorname{supp}(f^*{}_\sharp q) \cap (1 - \alpha, 1] = \varnothing$.

Because $\mathcal{D}_\triangle(f^*{}_\sharp p, f^*{}_\sharp q) \geq \mathbb{E}_{t \sim f^*{}_\sharp p}\big[d(t, \operatorname{supp}(f^*{}_\sharp q))\big] \geq \mathbb{P}_{f^*{}_\sharp p}(\{1\}) \cdot d(1, \operatorname{supp}(f^*{}_\sharp q)) \geq \mathbb{P}_{f^*{}_\sharp p}(\{1\}) \cdot \alpha > 0$, which contradicts with the given $\mathcal{D}_\triangle(f^*{}_\sharp p, f^*{}_\sharp q) = 0$, we have $\mathbb{E}_{x \sim p}[d(x, \operatorname{supp}(q))] = 0$.

2) Similarly, it can be shown $\mathbb{E}_{x \sim q}[d(x, \operatorname{supp}(p))] = 0$.

Thus, $\mathcal{D}_\triangle(f^*{}_\sharp p, f^*{}_\sharp q) = 0$ implies $\mathcal{D}_\triangle(p, q) = 0$.

### A.3 PROOF OF PROPOSITION 2.2

Consider a 1-dimensional Euclidean space $\mathbb{R}$. Let $\operatorname{supp}(p) = [-\frac{1}{2}, \frac{1}{2}] \cup [1, 2]$ with $p([-\frac{1}{2}, \frac{1}{2}]) = \frac{3}{4}$ and $p([1, 2]) = \frac{1}{4}$. Let $\operatorname{supp}(q) = [-2, -1] \cup [-\frac{1}{2}, \frac{1}{2}] \cup [1, 2]$ with $q([-2, -1]) = \frac{1}{4}, q([-\frac{1}{2}, \frac{1}{2}]) = \frac{1}{4}$ and $q([1, 2]) = \frac{1}{2}$. The supports of $p$ and $q$ consist of disjoint closed intervals, and we assume uniform distribution within each of these intervals, i.e. $p$ has density $p(x) = \frac{3}{4}, \forall x \in [-\frac{1}{2}, \frac{1}{2}]; p(x) = \frac{1}{4}, \forall x \in [1, 2]$ and $q$ has densitiy $q(x) = \frac{1}{4}, \forall x \in [-2, -1]; q(x) = \frac{1}{4}, \forall x \in [-\frac{1}{2}, \frac{1}{2}]; q(x) = \frac{1}{2}, \forall x \in [1, 2]$. Clearly, $\operatorname{supp}(p) \neq \operatorname{supp}(q)$.

The optimal dual Wasserstein discriminator $f_W^*$ is the maximizer of

$$\sup_{f:\operatorname{Lip}(f) \leq 1} \mathbb{E}_{x \sim p}[f(x)] - \mathbb{E}_{y \sim q}[f(y)].$$

Thus, $f_W^*$ is the maximizer of

$$\sup_{f:\operatorname{Lip}(f) \leq 1} \frac{1}{4}\left(3\int_{-\frac{1}{2}}^{\frac{1}{2}} f(x)dx + \int_1^2 f(x)dx - \int_{-2}^{-1} f(x)dx - \int_{-\frac{1}{2}}^{\frac{1}{2}} f(x)dx - 2\int_1^2 f(x)dx\right),$$

which simplifies to

$$\sup_{f:\operatorname{Lip}(f) \leq 1} \frac{1}{4}\left(-\int_{-2}^{-1} f(x)dx + 2\int_{-\frac{1}{2}}^{\frac{1}{2}} f(x)dx - \int_1^2 f(x)dx\right).$$

Since the optimization objective and the constraint are invariant to replacing the function $f(x)$ with its symmetric reflection $g(x) = f(-x)$, if $f'$ is a optimal solution, then there exists a symmetric

maximizer $f_W^*(x) = \frac{1}{2}f'(x) + \frac{1}{2}f'(-x)$, since $f_W^*(x) = f_W^*(-x)$ and $\mathrm{Lip}(f_W^*) \leq \mathrm{Lip}(f') \leq 1$. Thus, $\mathrm{supp}(f_{W\sharp}^*p) = \mathrm{supp}(f_{W\sharp}^\star q)$ as $f_W^*(x) = f_W^*(-x)$ for $x \in [1, 2]$.

Note that one can easily "extend" the above proof to discrete distributions, by replacing the disjoint segments $[-2, -1], [-\frac{1}{2}, \frac{1}{2}], [1, 2]$ with points $\{-1\}, \{0\}, \{1\}$.

## A.4 PROOF OF PROPOSITION 4.1

From (8), we have

$$\mathcal{D}_W^\infty(p, q) := \lim_{\beta \to \infty} \mathcal{D}_W^\beta(p, q) = \lim_{\beta \to \infty} \inf_{\gamma \in \Gamma_\beta(p,q)} \mathbb{E}_{(x,y) \sim \gamma}[d(x, y)],$$

where $\lim_{\beta \to \infty} \Gamma_\beta(p, q)$ is the set of all measures $\gamma$ on $\mathcal{X} \times \mathcal{X}$ such that $\int \gamma(x, y)dy = p(x), \forall x$ and $\int \gamma(x, y)dx \leq \lim_{\beta \to \infty}(1 + \beta)q(y), \forall y$.

The set of inequalities

$$\int \gamma(x, y)dx \leq \lim_{\beta \to \infty}(1 + \beta)q(y), \qquad \forall y$$

can be simplified to

$$\int \gamma(x, y)dx = 0, \qquad \forall y \text{ such that } q(y) = 0.$$

To put it together, we have

$$\mathcal{D}_W^\infty(p, q) = \inf_{\gamma \in \Gamma_\infty(p,q)} \mathbb{E}_{(x,y) \sim \gamma}[d(x, y)],$$

where $\Gamma_\infty(p, q)$ is the set of all measures $\gamma$ on $\mathcal{X} \times \mathcal{X}$ such that $\int \gamma(x, y)dy = p(x), \forall x$ and $\int \gamma(x, y)dx = 0, \forall y$ such that $q(y) = 0$. In other words, we seek the coupling $\gamma(x, y)$ which defines a transportation plan such that the total mass transported from given point $x$ is equal to $p(x)$, and the only constraint on the destination points $y$ is that no probability mass can be transported to points $y$ where $q(y) = 0$, i.e. $y \notin \mathrm{supp}(q)$.

Let $y^*(x)$ denote a function such that

$$y^*(x) \in \mathrm{supp}(q), \ \forall x; \qquad d(x, y^*(x)) = \inf_{y \in \mathrm{supp}(q)} d(x, y).$$

We can see that $\gamma^*$ given by

$$\gamma^*(x, y) = p(x)\delta(y - y^*(x)),$$

is the optimal coupling. Indeed, $\gamma^*$ satisfies the constraints $\gamma^* \in \Gamma_\infty$, and the cost of any other transportation cost $\gamma \in \Gamma_\infty$ is at least that of $\gamma^*$ (since $y^*(x)$ is defined as a closest point $y$ in $\mathrm{supp}(q)$ to a given point $x$).

Thus,

$$\mathcal{D}_W^\infty(p, q) = \inf_{\gamma \in \Gamma_\infty(p,q)} \mathbb{E}_{(x,y) \sim \gamma}[d(x, y)] = \mathbb{E}_{(x,y) \sim \gamma^*}[d(x, y)] = \mathbb{E}_{x \sim p}\left[\inf_{y \in \mathrm{supp}(q)} d(x, y)\right].$$

The last equation implies $\mathcal{D}_W^\infty(p, q) = \mathrm{SD}(p, q)$ ($\mathrm{SD}(\cdot, \cdot)$ is defined in (13)). Then,

$$\mathcal{D}_W^{\infty,\infty}(p, q) := \lim_{\beta_1, \beta_2 \to \infty} \mathcal{D}_W^{\beta_1, \beta_2}(p, q) = \mathcal{D}_W^\infty(p, q) + \mathcal{D}_W^\infty(q, p) = \mathrm{SD}(p, q) + \mathrm{SD}(q, p) = \mathcal{D}_\triangle(p, q).$$

## A.5 PROOF OF PROPOSITION 4.2

1. $\mathcal{D}_W(p, q) = 0$ implies $p = q$, which is equivalent to

$$\frac{p(x)}{q(x)} = 1, \qquad \forall x \in \mathrm{supp}(p) \cup \mathrm{supp}(q).$$

Then clearly, for all finite $\beta_1, \beta_2 > 0$ it satisfies

$$\frac{1}{1 + \beta_2} \leq \frac{p(x)}{q(x)} \leq 1 + \beta_1, \qquad \forall x \in \mathrm{supp}(p) \cup \mathrm{supp}(q). \tag{14}$$

Thus, $\mathcal{D}_W^{\beta_1, \beta_2}(p, q) = 0$ for all finite $\beta_1, \beta_2 > 0$.

2. $\mathcal{D}_W^{\beta_1,\beta_2}(p,q) = 0$ for some finite $\beta_1, \beta_2 > 0$ means that (14) is satisfied. This implies that $\forall x \in \operatorname{supp}(p)$, $x \in \operatorname{supp}(q)$ and $\forall x \in \operatorname{supp}(q)$, $x \in \operatorname{supp}(p)$, which makes $\operatorname{supp}(p) = \operatorname{supp}(q)$. Thus, $\mathcal{D}_\triangle(p,q) = 0$.

3. The converse of statements 1 and 2 are false:

   (a) For all finite $\beta_1, \beta_2 > 0$, let $\operatorname{supp}(p) = \operatorname{supp}(q) = \{x_1, x_2\}$. Let $p(x_1) = p(x_2) = 1/2$ and $q(x_1) = (1 + \beta')/2$ and $q(x_2) = (1 - \beta')/2$ where

$$\beta' = \min\left(\beta_2, 1 - \frac{1}{1 + \beta_1}\right).$$

   Then, it can be easily checked that (14) is satisfied, which makes $\mathcal{D}_W^{\beta_1,\beta_2}(p,q) = 0$. However, since $\beta' \neq 0$, $p \neq q$ and thus $\mathcal{D}_W(p,q) \neq 0$.

   (b) Similar to (a), let $\operatorname{supp}(p) = \operatorname{supp}(q) = \{x_1, x_2\}$. Let $p(x_1) = q(x_2) = \varepsilon$ and $p(x_2) = q(x_1) = 1 - \varepsilon$ for some $\varepsilon > 0$. Since $\operatorname{supp}(p) = \operatorname{supp}(q)$, $\mathcal{D}_\triangle(p,q) = 0$. However,

$$\lim_{\varepsilon \downarrow 0} \frac{p(x_1)}{q(x_1)} = \lim_{\varepsilon \downarrow 0} \frac{\varepsilon}{1 - \varepsilon} = 0,$$

   and, thus, for any finite $\beta_2 > 0$ we can choose $\varepsilon > 0$ such that

$$\frac{p(x_1)}{q(x_1)} = \frac{\varepsilon}{1 - \varepsilon} < \frac{1}{1 + \beta_2}.$$

   Therefore, (14) is not satisfied and $\mathcal{D}_W^{\beta_1,\beta_2}(p,q) \neq 0$.

## A.6  PROOF OF PROPOSITION 4.3

Using (2), we first establish a connection between the pushforward distributions $f^*_\sharp p$ and $f^*_\sharp q$.

**Proposition A.1.** *Let $f^*$ be the optimal log-loss discriminator (2) between $p$ and $q$. Then,*

$$\frac{[f^*_\sharp p](t)}{[f^*_\sharp p](t) + [f^*_\sharp q](t)} = t, \qquad \forall\, t \in \operatorname{supp}(f^*_\sharp p) \cup \operatorname{supp}(f^*_\sharp q). \tag{15}$$

**Proof.** For any point $t \in \operatorname{supp}(f^*_\sharp p) \cup \operatorname{supp}(f^*_\sharp q)$, the values of the densities

$$[f^*_\sharp p](t) = \lim_{\varepsilon \downarrow 0} \frac{\mathbb{P}_p\left(\{x \mid t - \varepsilon < f^*(x) < t + \varepsilon\}\right)}{2\varepsilon} = \lim_{\varepsilon \downarrow 0} \frac{\int_{\{x \mid t - \varepsilon < f^*(x) < t+\varepsilon\}} p(x)\, dx}{2\varepsilon},$$

$$[f^*_\sharp q](t) = \lim_{\varepsilon \downarrow 0} \frac{\mathbb{P}_q\left(\{x \mid t - \varepsilon < f^*(x) < t + \varepsilon\}\right)}{2\varepsilon} = \lim_{\varepsilon \downarrow 0} \frac{\int_{\{x \mid t - \varepsilon < f^*(x) < t+\varepsilon\}} q(x)\, dx}{2\varepsilon}.$$

Note that for all $x : t - \varepsilon < f^*(x) < t + \varepsilon$ we have

$$t - \varepsilon < \frac{p(x)}{p(x) + q(x)} < t + \varepsilon,$$

which implies

$$(t - \varepsilon)(p(x) + q(x)) < p(x) < (t + \varepsilon)(p(x) + q(x)).$$

Since these inequalities hold for all $x : t - \varepsilon < f^*(x) < t + \varepsilon$, the similar relationship holds for the integrals:

$$(t - \varepsilon) \int_{\{x \mid t-\varepsilon<f^*(x)<t+\varepsilon\}} (p(x) + q(x))\, dx$$

$$< \int_{\{x \mid t-\varepsilon<f^*(x)<t+\varepsilon\}} p(x)\, dx <$$

$$(t + \varepsilon) \int_{\{x \mid t-\varepsilon<f^*(x)<t+\varepsilon\}} (p(x) + q(x))\, dx.$$

The ratio $[f^*{}_\sharp p](t)/([f^*{}_\sharp p](t) + [f^*{}_\sharp q](t))$ can be expressed as

$$\frac{[f^*{}_\sharp p](t)}{[f^*{}_\sharp p](t) + [f^*{}_\sharp q](t)} = \lim_{\varepsilon \downarrow 0} \frac{\int_{\{x|t-\varepsilon<f^*(x)<t+\varepsilon\}} p(x)\,dx}{\int_{\{x|t-\varepsilon<f^*(x)<t+\varepsilon\}} (p(x)+q(x))\,dx}.$$

Using the inequality above we observe that

$$t - \varepsilon < \frac{\int_{\{x|t-\varepsilon<f^*(x)<t+\varepsilon\}} p(x)\,dx}{\int_{\{x|t-\varepsilon<f^*(x)<t+\varepsilon\}} (p(x)+q(x))\,dx} < t + \varepsilon,$$

for all $\varepsilon > 0$, and taking the limit $\varepsilon \downarrow 0$ we obtain

$$\frac{[f^*{}_\sharp p](t)}{[f^*{}_\sharp p](t) + [f^*{}_\sharp q](t)} = t.$$

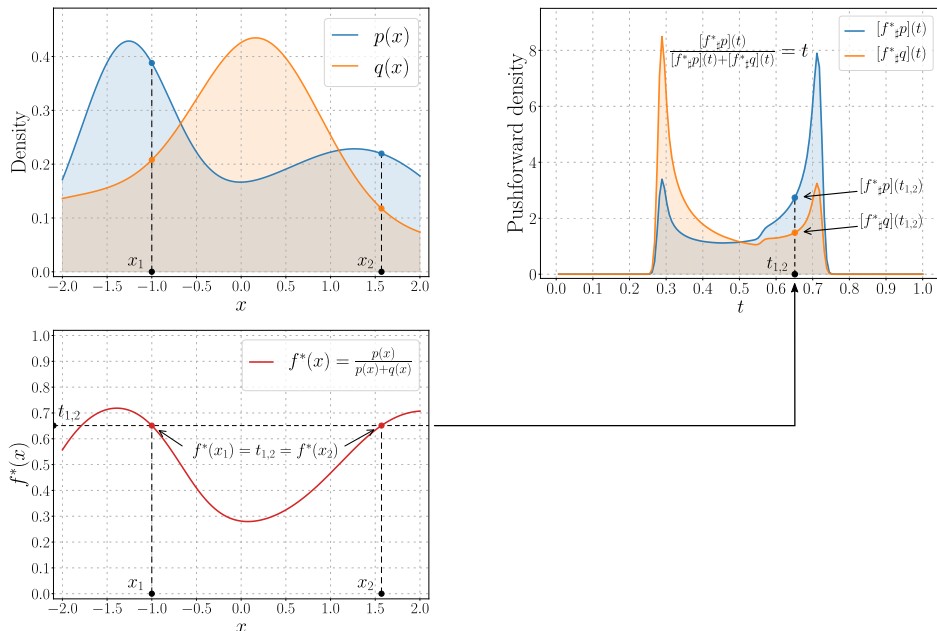

Figure A.1: Visual illustration of the statement of Proposition A.1. The top-left panel shows two example PDFs $p(x)$, $q(x)$ on closed interval $[-2, 2]$. The bottom-left panel shows the optimal discriminator function $f^*(x) = p(x)/(p(x) + q(x))$ as a function of $x$ on $[-2, 2]$. The top-right panel shows the PDFs $[f^*{}_\sharp p](t)$, $[f^*{}_\sharp q](t)$ of the pushforward distributions $f^*{}_\sharp p$, $f^*{}_\sharp q$ induced by the discriminator mapping $f^*$. $f^*$ maps $[-2, 2]$ to $[0, 1]$ and $[f^*{}_\sharp p]$, $[f^*{}_\sharp q]$ are defined on $[0, 1]$. Consider point $x_1 \in [-2, 2]$. The value $f^*(x_1)$ characterizes the ratio of densities $p(x_1)/(p(x_1) + q(x_1))$ at $x_1$. For another point $x_2$ mapped to the same value $f^*(x_2) = f^*(x_1) = t_{1,2}$, the ratio of densities $p(x_2)/(p(x_2) + q(x_2))$ is the same as $p(x_1)/(p(x_1) + q(x_1))$. All points $x$ mapped to $t_{1,2}$ share the same ratio of the densities $p(x)/(p(x) + q(x))$. This fact implies that the ratio of the pushforward densities $[f^*{}_\sharp p](t_{1,2})/([f^*{}_\sharp p](t_{1,2}) + [f^*{}_\sharp q](t_{1,2}))$ at $t_{1,2}$ must be the same as the ratio of densities $p(x_1)/(p(x_1) + q(x_1)) = t_{1,2}$ at $x_1$ (or $x_2$). The pushforward PDFs $[f^*{}_\sharp q](t)$, $[f^*{}_\sharp q](t)$ satisfy property $[f^*{}_\sharp p](t)/([f^*{}_\sharp p](t) + [f^*{}_\sharp q](t)) = t$ for all $t \in \text{supp}(f^*{}_\sharp p) \cup \text{supp}(f^*{}_\sharp q)$.

**Comment.** Intuitively this proposition states the following. If for some $x \in \mathcal{X}$ we have $f^*(x) = t \in [0, 1]$, $t$ directly corresponds to the ratio of densities not only in the original space $t = p(x)/(p(x) + q(x))$, but also in the 1D discriminator output space $t = [f^*{}_\sharp p](t)/([f^*{}_\sharp p](t) + [f^*{}_\sharp q](t))$.

We also provide an intuitive example in Figure A.1.

**Proof of Proposition 4.3, statement #1.**

$\implies$: If $\mathcal{D}_W(p, q) = 0$ then $p = q$, then $f^*{}_\sharp p = f^*{}_\sharp q$. Thus, $\mathcal{D}_W(f^*{}_\sharp p, f^*{}_\sharp q) = 0$.

$\Longleftarrow$: If $\mathcal{D}_W(f^*_\sharp p, f^*_\sharp q) = 0$, then $f^*_\sharp p = f^*_\sharp q$.

Consider probability of event $\left\{ t \mid t > \frac{1}{2} \right\}$ under distribution $f^*_\sharp p$.

$$\mathbb{P}_{f^*_\sharp p}\left(\left\{ t \,\Big|\, t > \frac{1}{2} \right\}\right) = \int \mathbb{I}\left[ f^*(x) > \frac{1}{2} \right] p(x)\, dx,$$

where $\mathbb{I}[\cdot]$ is the indicator function ($\mathbb{I}[c]$ equal to 1 when the condition $c$ is satisfied, and equal to 0 otherwise). For all $x : p(x) > 0$, we have that $f^*(x) = \frac{p(x)}{p(x)+q(x)}$ and $p(x) + q(x) > 0$. Therefore, the expression above can be re-written as

$$\mathbb{P}_{f^*_\sharp p}\left(\left\{ t \,\Big|\, t > \frac{1}{2} \right\}\right) = \int \mathbb{I}\left[ \frac{p(x)}{p(x)+q(x)} > \frac{1}{2} \right] p(x)\, dx = \int \mathbb{I}[p(x) - q(x) > 0] p(x)\, dx.$$

Similarly, the probability of event $\left\{ t \mid t > \frac{1}{2} \right\}$ under distribution $f^*_\sharp q$ is

$$\mathbb{P}_{f^*_\sharp q}\left(\left\{ t \,\Big|\, t > \frac{1}{2} \right\}\right) = \int \mathbb{I}[p(x) - q(x) > 0] q(x)\, dx.$$

$f^*_\sharp p = f^*_\sharp q$ implies that

$$\mathbb{P}_{f^*_\sharp p}\left(\left\{ t \,\Big|\, t > \frac{1}{2} \right\}\right) = \mathbb{P}_{f^*_\sharp q}\left(\left\{ t \,\Big|\, t > \frac{1}{2} \right\}\right),$$

or equivalently

$$\int \mathbb{I}[p(x) - q(x) > 0](p(x) - q(x))\, dx = 0.$$

Note, that the function $\mathbb{I}[p(x) - q(x) > 0](p(x) - q(x))$ is non-negative for any $x$. This means that the integral can be zero only if the function is zero everywhere implying that for any $x$ either $\mathbb{I}[p(x) - q(x) > 0] = 0$ or $p(x) - q(x) = 0$. In other words,

$$p(x) \le q(x), \quad \forall\, x.$$

Using the fact the both densities $p(x)$ and $q(x)$ must sum up to 1, we conclude that $p = q$ and $\mathcal{D}_W(p, q) = 0$.

**Proof of Proposition 4.3, statement #2.**

Note that by (2) and (15), we have

$$f^*(x) = \frac{p(x)}{p(x)+q(x)} = t = \frac{[f^*_\sharp p](t)}{[f^*_\sharp p](t) + [f^*_\sharp q](t)}, \qquad \forall x \in \mathrm{supp}(p) \cup \mathrm{supp}(q).$$

$\Longrightarrow$: Suppose $\mathcal{D}_W^{\beta_1, \beta_2}(p, q) = 0$. Then $\mathrm{supp}(p) = \mathrm{supp}(q) = S$ (by Proposition 4.2) and $\mathrm{supp}(f^*_\sharp p) = \mathrm{supp}(f^*_\sharp q) = T$ by (Theorem 2.1). Moreover, $\mathcal{D}_W^{\beta_1, \beta_2}(p, q) = 0$ implies

$$\frac{1}{1+\beta_2} \le \frac{p(x)}{q(x)} \le 1 + \beta_1, \qquad \forall x \in S.$$

Since

$$f^*(x) = \frac{p(x)}{p(x)+q(x)} = \frac{\frac{p(x)}{q(x)}}{1 + \frac{p(x)}{q(x)}}, \qquad \forall x \in S,$$

the inequalities above are equivalent to

$$\frac{1}{2+\beta_2} \le f^*(x) \le \frac{1+\beta_1}{2+\beta_1}, \qquad \forall x \in S.$$

Combined with Proposition A.1, the above implies that

$$\frac{1}{2 + \beta_2} \leq \frac{[f^*{}_\sharp p](t)}{[f^*{}_\sharp p](t) + [f^*{}_\sharp q](t)} \leq \frac{1 + \beta_1}{2 + \beta_1}, \qquad \forall t \in T,$$

or equivalently

$$\frac{1}{1 + \beta_2} \leq \frac{[f^*{}_\sharp p](t)}{[f^*{}_\sharp q](t)} \leq 1 + \beta_1, \qquad \forall t \in T.$$

Therefore, $\mathcal{D}_W^{\beta_1, \beta_2}(f^*{}_\sharp p, f^*{}_\sharp q) = 0$.

---

$\Longleftarrow$: similarly, when $\mathcal{D}_W^{\beta_1, \beta_2}(f^*{}_\sharp p, f^*{}_\sharp q) = 0$,

$$\mathrm{supp}(f^*{}_\sharp p) = \mathrm{supp}(f^*{}_\sharp q) = T \qquad \Longrightarrow \qquad \mathrm{supp}(p) = \mathrm{supp}(q) = S.$$

Moreover,

$$\frac{1}{1 + \beta_2} \leq \frac{[f^*{}_\sharp p](t)}{[f^*{}_\sharp q](t)} \leq 1 + \beta_1, \qquad \forall t \in T,$$

$$\Downarrow$$

$$\frac{1}{2 + \beta_2} \leq \frac{[f^*{}_\sharp p](t)}{[f^*{}_\sharp p](t) + [f^*{}_\sharp q](t)} \leq \frac{1 + \beta_1}{2 + \beta_2}, \qquad \forall t \in T$$

$$\Downarrow$$

$$\frac{1}{2 + \beta_2} \leq f^*(x) \leq \frac{1 + \beta_1}{2 + \beta_1}, \qquad \forall x \in S,$$

$$\Downarrow$$

$$\frac{1}{1 + \beta_2} \leq \frac{p(x)}{q(x)} \leq 1 + \beta_1, \qquad \forall x \in S.$$

Therefore, $\mathcal{D}_W^{\beta_1, \beta_2}(p, q) = 0$.

## B    A COMMENT ON "SLICED" SSD DIVERGENCE

Recent works (Deshpande et al., 2018; 2019) have proposed to perform optimal transport (OT)-based distribution alignment by reducing the OT problem (7) in the original, potentially high-dimensional space, to that between 1D distributions. Specifically, Deshpande et al. (2018) consider the sliced Wasserstein distance (Rabin et al., 2011):

$$\mathcal{D}_{SW}(p, q) = \int_{\mathbb{S}^{n-1}} \mathcal{D}_W(f^\theta{}_\sharp p, f^\theta{}_\sharp q) \, d\theta, \tag{16}$$

where $\mathbb{S}^{n-1} = \{\theta \in \mathbb{R}^n \mid \|\theta\| = 1\}$ is a unit sphere in $\mathbb{R}^n$, and $f^\theta$ is a 1D linear projection $f^\theta(x) = \langle \theta, x \rangle$. It is known that $\mathcal{D}_{SW}$ is a valid distribution divergence: for any $p \neq q$ there exists a linear slicing function $f^\theta$, $\theta \in \mathbb{S}^{n-1}$ which identifies difference in the distributions, i.e. $f^\theta{}_\sharp p \neq f^\theta{}_\sharp q$ (Cramér–Wold theorem). By reducing the original OT problem to that in a 1D space, Wu et al. (2019a) and Deshpande et al. (2019) develop efficient practical methods for distribution alignment based on fast algorithms for the 1D OT problem.

Unfortunately, the straight-forward extension of SSD divergence (1) to a 1D linearly sliced version does not provide a valid support divergence.

**Proposition B.1.** *There exist two distributions $p$ and $q$ in $\mathcal{P}$, such that $\mathrm{supp}(p) \neq \mathrm{supp}(q)$ but $\mathrm{supp}(f^\theta{}_\sharp p) = \mathrm{supp}(f^\theta{}_\sharp q)$, $\forall f^\theta(x) = \langle \theta, x \rangle$ with $\theta \in \mathbb{S}^{n-1}$.*

*Proof.* Consider a 2-dimensional Euclidean space $\mathbb{R}^2$ and let $\mathrm{supp}(p) = \{(x, y) | x^2 + y^2 \leq 2\}$ and $\mathrm{supp}(q) = \{(x, y) | 1 \leq x^2 + y^2 \leq 2\}$. Then, $\forall f^\theta(x) = \langle \theta, x \rangle$ with $\theta \in \mathbb{S}^1$,

$$\mathrm{supp}(f^\theta{}_\sharp p) = \mathrm{supp}(f^\theta{}_\sharp q) = [-2, 2].$$

This counterexample is shown in Figure B.1. $\qquad \square$

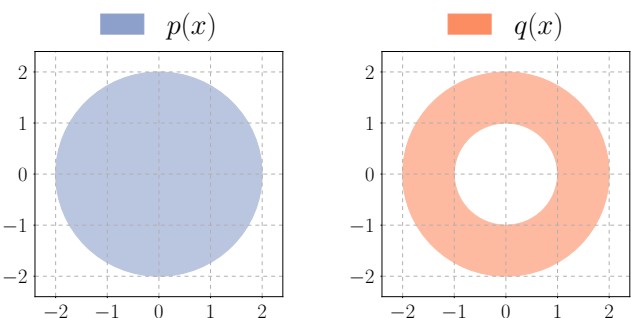

Figure B.1: Visualization of example distributions for Proposition B.1

## C DISCUSSION OF "SOFT" AND "HARD" ASSIGNMENTS WITH 1D DISCRETE DISTRIBUTIONS

In Section 4.2 we considered the "soft-assignment" relaxed OT problem (11) and claimed that for integer $\beta$, the set of minimizers of (11) must contain a "hard-assignment" transportation plan, meaning $\gamma_{ij} \in \{0, 1\}, \forall i, j$. Below we justify this claim.

Note that for $\beta = 0$ the OT problem (11) is the standard OT problem for Wasserstein-1 distance (10), since the inequality constraints $\sum_{i=1}^{m} \gamma_{ij} \leq 1, \forall j$ can only be satisfied as equalities. For this problem, it is known (e.g. see Peyré et al. (2019) Proposition 2.1) that the set of optimal "soft-assignment" contains a "hard-assignment" represented by a normalized permutation matrix. This fact can be proven using the Birkhoff–von Neumann theorem. The Birkhoff–von Neumann theorem states that the set of doubly stochastic matrices

$$P \in \mathbb{R}^{n \times n} : \qquad P_{ij} \geq 0, \forall\, i, j, \qquad \sum_{j=1}^{n} P_{ij} = 1, \forall\, i, \qquad \sum_{i=1}^{n} P_{ij} = 1, \forall\, j$$

is exactly the set of all finite convex combinations of permutation matrices. In the context of the linear program (11) with $\beta = 0$, the Birkhoff–von Neumann theorem means that all extreme points of the polyhedron $\Gamma_\beta(o^p, o^q)$ are hard-assignment matrices. Therefore, by the fundamental theorem of linear programming (Bertsimas & Tsitsiklis, 1997), the minimum of the objective is reached at a "hard-assignment" matrix.

We argue that a similar result holds for the case of integer $\beta > 0$. In this case, the matrices in $\Gamma_\beta(o^p, o^q)$ can not be associated with the doubly stochastic matrices, since constraints on of the marginals of $\gamma$ are relaxed to inequality constraints. Because of that, the Birkhoff–von Neumann theorem can not be applied. However, Budish et al. (2009) provide a generalization of the Birkhoff–von Neumann theorem (Theorem 1 in (Budish et al., 2009)) which applies to the cases where the equality constraints are replaced with integer-valued inequality constraints (recall that we consider integer $\beta$). Using this generalized result, our claim can be proven by performing the following steps.

Clearly, the polyhedron $\Gamma_\beta(o^p, o^q)$ contains all "hard-assignment" matrices and all their finite convex combinations. The result proven in (Budish et al., 2009) implies that each element of $\Gamma_\beta(o^p, o^q)$ can be represented as a finite convex combination of "hard-assignment" matrices. Thus, the polyhedron $\Gamma_\beta(o^p, o^q)$ is exactly the set of all finite convex combinations of "hard-assignment" matrices and all extreme points of the polyhedron are "hard-assignment" matrices. Finally, by analogy with the case of $\beta = 0$, we invoke the fundamental theorem of the linear programming and conclude that the minimum of the objective (11) is reached at $\gamma$ corresponding to a "hard-assignment" matrix.

## D EXPERIMENT DETAILS

### D.1 USPS TO MNIST EXPERIMENT SPECIFICATIONS

We use USPS (Hull, 1994) and MNIST (LeCun et al., 1998) datasets for this adaptation problem.

Following Tachet des Combes et al. (2020) we use LeNet-like (LeCun et al., 1998) architecture for the feature extractor with the 500-dimensional feature representation. The classifier consists of a single linear layer. The discriminator is implemented by a 3-layer MLP with 512 hidden units and leaky-ReLU activation.

We train all methods for $65\,000$ steps with batch size $64$. We train the feature extractor, the classifier, and the discriminator with SGD (learning rate $0.02$, momentum $0.9$, weight decay $5 \cdot 10^{-4}$). We perform a single discriminator update per 1 update of the feature extractor and the classifier. After the first $30\,000$ steps we linearly anneal the feature extractor's and classifier's learning rates for $30\,000$ steps to the final value $2 \cdot 10^{-5}$.

The feature extractor's loss is given by a weighted combination of the cross-entropy classification loss on the labeled source example and a domain alignment loss computed from the discriminator's signal (recall that different method use different forms of the alignment loss). The weight for the classification term is constant and set to $\lambda_{\mathrm{cls}} = 1$. We introduce schedule for the alignment weight $\lambda_{\mathrm{align}}$. For all alignment methods we linearly increase $\lambda_{\mathrm{align}}$ from 0 to 1.0 during the first 10000 steps.

For ASA we use history buffers of size 1000.

### D.2 STL TO CIFAR EXPERIMENT SPECIFICATIONS

We use STL (Coates et al., 2011) and CIFAR-10 (Krizhevsky, 2009) for this adaptation task. STL and CIFAR-10 are both 10-class classification problems. There are 9 common classes between the two datasets. Following Shu et al. (2018) we create a 9-class classification problem by selecting the subsets of examples of the 9 common classes.

For the feature extractor, we adapt the deep CNN architecture of Shu et al. (2018). The feature representation is a 192-dimensional vector. The classifier consists of a single linear layer. The discriminator is implemented by a 3-layer MLP with 512 hidden units and leaky-ReLU activation.

We train all methods for $40\,000$ steps with batch size $64$. We train the feature extractor, the classifier, and the discriminator with ADAM (Kingma & Ba, 2014) (learning rate $0.001$, $\beta_1 = 0.5$, $\beta_2 = 0.999$, no weight decay). We perform a single discriminator update per 1 update of the feature extractor and the classifier.

The weight for the classification loss term is constant and set to $\lambda_{\mathrm{cls}} = 1$. For all alignment methods we use constant alignment weight $\lambda_{\mathrm{align}} = 0.1$.

For ASA we use history buffers of size 1000.

**Conditional entropy loss.** Following (Shu et al., 2018) we use auxiliary conditional entropy loss on target examples for domain adaptation methods. For a classifier $C^{\phi} : \mathcal{Z} \to \mathcal{Y}$ and a feature extractor $F^{\theta} : \mathcal{X} \to \mathcal{Z}$ where classifier outputs the distribution over class labels $\{1, \ldots, K\}$

$$C^{\phi}(z) \in \mathbb{R}^K : \quad \left[C^{\phi}(z)\right]_k \geq 0, \quad \sum_{k=1}^{K} \left[C^{\phi}(z)\right]_k = 1,$$

the conditional entropy loss on target examples $\{x_i^q\}_{i=1}^{N_q}$ is given by

$$\mathcal{L}_{\mathrm{ent}} = \lambda_{\mathrm{ent}} \cdot \frac{1}{N^q} \sum_{i=1}^{N_q} \left( -\sum_{k=1}^{K} \left[C^{\phi}(F^{\theta}(x_i^q))\right]_k \log \left[C^{\phi}(F^{\theta}(x_i^q))\right]_k \right). \tag{17}$$

$\lambda_{\mathrm{ent}}$ is the weight of the conditional entropy loss in the total training objective. This loss acts as an additional regularization of the embeddings of the unlabeled target examples: minimization of the conditional entropy pushes target embeddings away from the classifier's decision boundary.

For all domain adapation methods we use the conditional entropy loss (17) on target examples with the weight $\lambda_{\mathrm{ent}} = 0.1$.

## D.3 EXTENDED EXPERIMENTAL RESULTS ON STL TO CIFAR

**Effect of conditional entropy loss.** In order to quantify the improvements of the support alignment objective and the conditional entropy objective in separation, we conduct an ablation study. In addition to the results reported in Section 5, we evaluate all domain adaptation methods (except VADA which uses the conditional entropy in the original implementation) on STL→CIFAR task without the conditional entropy loss ($\lambda_{\mathrm{ent}} = 0$). The results of the ablation study are presented in Table D.1. We observe that the effect of the auxiliary conditional entropy is essentially the same for all methods across all imbalance levels: with $\lambda_{\mathrm{ent}} = 0.1$ the accuracy either improves (especially the average class accuracy) or roughly stays on the same level. The relative ranking of distribution alignment, relaxed distribution alignment, and support alignment methods is the same with both $\lambda_{\mathrm{ent}} = 0$ and $\lambda_{\mathrm{ent}} = 0.1$. The results demonstrate that the benefits of support alignment approach and conditional entropy are orthogonal.

Table D.1: Results of ablation experiments of the effect of auxiliary conditional entropy loss on STL→CIFAR data. Same setup and reporting metrics as Table 1.

| Algorithm | $\lambda_{\mathrm{ent}}$ | $\alpha = 0.0$ | | $\alpha = 1.0$ | | $\alpha = 1.5$ | | $\alpha = 2.0$ | |
| | | average | min | average | min | average | min | average | min |
|---|---|---|---|---|---|---|---|---|---|
| DANN | 0.0 | $74.6^{75.1}_{74.1}$ | $51.5^{55.0}_{49.9}$ | $68.4^{69.2}_{67.0}$ | $43.2^{43.7}_{41.2}$ | $65.7^{65.9}_{62.8}$ | $35.5^{36.2}_{29.6}$ | $62.5^{64.6}_{60.0}$ | $27.5^{27.5}_{25.7}$ |
| DANN | 0.1 | $75.3^{75.4}_{74.9}$ | $54.6^{56.6}_{54.2}$ | $69.9^{70.1}_{68.6}$ | $44.8^{45.1}_{40.7}$ | $64.9^{67.1}_{63.7}$ | $34.9^{36.8}_{33.9}$ | $63.3^{64.8}_{57.4}$ | $27.0^{28.5}_{21.2}$ |
| IWDAN | 0.0 | $70.4^{70.7}_{70.2}$ | $47.2^{48.0}_{46.8}$ | $68.6^{68.8}_{68.4}$ | $43.6^{46.3}_{43.2}$ | $66.7^{67.9}_{66.0}$ | $44.7^{46.2}_{43.3}$ | $63.9^{66.1}_{62.9}$ | $36.5^{37.3}_{32.7}$ |
| IWDAN | 0.1 | $69.9^{70.7}_{69.9}$ | $50.5^{50.6}_{47.9}$ | $68.7^{69.1}_{68.6}$ | $45.8^{50.5}_{44.8}$ | $67.1^{67.3}_{65.9}$ | $44.7^{44.8}_{40.4}$ | $64.4^{64.9}_{63.6}$ | $36.8^{37.9}_{34.5}$ |
| IWCDAN | 0.0 | $70.1^{70.8}_{70.0}$ | $50.5^{50.8}_{49.1}$ | $68.6^{69.4}_{68.2}$ | $44.2^{45.8}_{41.2}$ | $66.0^{66.0}_{65.9}$ | $45.0^{47.8}_{43.7}$ | $63.8^{64.1}_{62.3}$ | $37.3^{37.7}_{33.6}$ |
| IWCDAN | 0.1 | $70.1^{70.2}_{70.1}$ | $47.8^{49.3}_{42.4}$ | $69.4^{69.4}_{69.1}$ | $47.1^{51.3}_{46.3}$ | $66.1^{67.2}_{65.0}$ | $39.9^{40.8}_{37.7}$ | $64.5^{65.1}_{63.9}$ | $37.0^{40.2}_{35.5}$ |
| sDANN-4 | 0.0 | $69.4^{70.0}_{68.8}$ | $46.5^{49.7}_{45.1}$ | $69.6^{69.7}_{69.3}$ | $49.1^{49.2}_{47.4}$ | $68.0^{68.6}_{67.8}$ | $48.2^{48.8}_{42.6}$ | $66.3^{66.4}_{64.2}$ | $40.7^{42.9}_{36.6}$ |
| sDANN-4 | 0.1 | $71.8^{72.1}_{71.7}$ | $52.1^{52.8}_{52.1}$ | $71.1^{71.7}_{70.4}$ | $49.9^{51.8}_{48.1}$ | $69.4^{70.0}_{68.7}$ | $48.6^{49.0}_{43.5}$ | $66.4^{67.9}_{66.2}$ | $39.0^{47.1}_{33.6}$ |
| ASA-sq | 0.0 | $69.9^{70.3}_{69.9}$ | $48.0^{50.1}_{46.6}$ | $68.8^{68.9}_{68.6}$ | $47.3^{49.3}_{45.3}$ | $68.1^{68.7}_{67.2}$ | $45.4^{47.8}_{45.2}$ | $65.7^{66.4}_{65.6}$ | $43.6^{45.0}_{41.3}$ |
| ASA-sq | 0.1 | $71.7^{71.9}_{71.7}$ | $52.9^{53.4}_{46.7}$ | $70.7^{71.0}_{70.4}$ | $51.6^{52.7}_{46.8}$ | $69.2^{69.3}_{69.2}$ | $45.6^{52.0}_{43.3}$ | $68.1^{68.2}_{67.2}$ | $44.7^{45.9}_{39.8}$ |
| ASA-abs | 0.0 | $69.8^{70.0}_{68.9}$ | $45.7^{48.0}_{45.4}$ | $68.4^{68.6}_{68.4}$ | $44.3^{46.8}_{44.0}$ | $67.9^{68.1}_{67.0}$ | $46.6^{48.4}_{40.4}$ | $66.3^{66.9}_{65.7}$ | $41.6^{44.9}_{40.3}$ |
| ASA-abs | 0.1 | $71.6^{71.7}_{71.2}$ | $49.0^{53.5}_{48.4}$ | $70.9^{71.0}_{70.8}$ | $49.2^{50.0}_{47.3}$ | $69.6^{69.9}_{69.6}$ | $43.2^{49.5}_{42.1}$ | $67.8^{68.2}_{66.6}$ | $40.9^{49.0}_{35.4}$ |

**Comparison with optimal transport based baselines.**

We provide additional experimental results comparing our method with OT-based methods for domain adaptation. We implement two OT-based methods which we describe below.

- The first method is a variant of the max-sliced Wasserstein distance (which was proposed for GAN training by Deshpande et al. (2019)) for domain adaptation. In the table below we refer to this method as DANN-OT. In our implementation DANN-OT minimizes the Wasserstein distance between the pushforward distributions $g^*_\sharp p$, $g^*_\sharp q$ induced by the optimal log-loss discriminator $g^*$ (4). As discussed in Section 4.2 (paragraph "Distribution alignment") the computation of the Wasserstein distance between 1D distributions can be implemented efficiently via sorting.

- The second method is an OT-based variant of DANN which uses a dual Wasserstein discriminator instead of the log-loss discriminator. In the table below we refer to this method as DANN-WGP. This method minimizes the Wasserstein distance in its dual Kantorovich form. We train the discriminator with the Wasserstein dual objective and a gradient penalty proposed to enforce Lipshitz-norm constraint (Gulrajani et al., 2017).

We present the evaluation results of the OT-based methods on STL→CIFAR domain adaptation task in the Table D.2. Note that the OT-based methods aim to enforce distribution alignment constraints. We observe that the OT-based methods follow the same trend as DANN: they deliver improved accuracy compared to No DA in the balanced setting, but suffer in the imbalanced settings ($\alpha > 0$) due to their distribution alignment nature.

We would also like to make a comment on OT-based relaxed distribution alignment. Wu et al. (2019b) propose method WDANN-$\beta$ which minimizes the dual form of the asymmetrically-relaxed

Table D.2: Results of comparison with optimal transport based methods on STL→CIFAR data. Same setup and reporting metrics as Table 1.

| Algorithm | $\alpha = 0.0$ | | $\alpha = 1.0$ | | $\alpha = 1.5$ | | $\alpha = 2.0$ | |
|---|---|---|---|---|---|---|---|---|
| | average | min | average | min | average | min | average | min |
| No DA | $69.9^{70.0}_{69.8}$ | $49.8^{50.6}_{45.3}$ | $68.8^{69.3}_{68.3}$ | $47.2^{48.2}_{45.3}$ | $66.8^{67.2}_{66.4}$ | $46.0^{47.0}_{45.8}$ | $65.8^{66.7}_{64.8}$ | $43.7^{44.6}_{41.6}$ |
| DANN | $75.3^{75.4}_{74.9}$ | $54.6^{56.6}_{54.2}$ | $69.9^{70.1}_{68.6}$ | $44.8^{45.1}_{40.7}$ | $64.9^{67.1}_{63.7}$ | $34.9^{36.8}_{33.9}$ | $63.3^{64.8}_{57.4}$ | $27.0^{28.5}_{21.2}$ |
| DANN-OT | $76.0^{76.0}_{75.8}$ | $55.2^{55.5}_{54.3}$ | $67.7^{68.9}_{67.1}$ | $43.0^{43.7}_{36.5}$ | $64.5^{65.1}_{60.9}$ | $34.4^{34.6}_{29.3}$ | $61.3^{62.0}_{54.4}$ | $24.3^{25.5}_{23.2}$ |
| DANN-WGP | $74.8^{75.1}_{74.7}$ | $53.5^{54.4}_{53.3}$ | $67.7^{67.9}_{65.3}$ | $38.6^{41.0}_{34.4}$ | $63.3^{63.4}_{57.1}$ | $27.0^{32.4}_{26.3}$ | $59.0^{61.8}_{54.3}$ | $21.9^{22.5}_{18.6}$ |
| sDANN-4 | $71.8^{72.1}_{71.7}$ | $52.1^{52.8}_{52.1}$ | $71.1^{71.7}_{70.4}$ | $49.9^{51.8}_{48.1}$ | $69.4^{70.0}_{68.7}$ | $48.6^{49.0}_{43.5}$ | $66.4^{67.9}_{66.2}$ | $39.0^{47.1}_{33.6}$ |
| ASA-sq | $71.7^{71.9}_{71.7}$ | $52.9^{53.4}_{46.7}$ | $70.7^{71.0}_{70.4}$ | $51.6^{52.7}_{46.8}$ | $69.2^{69.3}_{69.2}$ | $45.6^{52.0}_{43.3}$ | $68.1^{68.2}_{67.2}$ | $44.7^{45.9}_{39.8}$ |
| ASA-abs | $71.6^{71.7}_{71.2}$ | $49.0^{53.5}_{48.4}$ | $70.9^{71.0}_{70.8}$ | $49.2^{50.0}_{47.3}$ | $69.6^{69.9}_{69.6}$ | $43.2^{49.5}_{42.1}$ | $67.8^{68.2}_{66.6}$ | $40.9^{49.0}_{35.4}$ |

Wasserstein distance. However, they observe that sDANN-$\beta$ outperforms WDANN-$\beta$ in experiments. Hence, we use sDANN-$\beta$ as a relaxed distribution alignment baseline in our experiments.

**Effect of alignment weight.** We provide additional experimental results comparing ASA with DANN and VADA across different values of the alignment loss weight $\lambda_{\text{align}}$ on STL→CIFAR task. The results are shown in Table D.3. DANN with a higher alignment weight $\lambda_{\text{align}} = 1.0$ performs better in the balanced ($\alpha = 0$) setting and worse in the imbalanced ($\alpha > 0$) setting compared to a lower weight $\lambda_{\text{align}} = 0.1$, as the distribution alignment constraint is enforced stricter. VADA optimizes a combination of distribution alignment + VAT (virtual adversarial training) objectives (Shu et al., 2018), and we observe the same trend: with lower alignment weight $\lambda_{\text{align}} = 0.01$, VADA performs worse in the balanced setting and better in the imbalanced setting compared to a higher weight $\lambda_{\text{align}} = 0.1$. Weight $\lambda_{\text{align}} = 0.1$ is a middle ground between having poor performance in the imbalanced setting ($\lambda_{\text{align}} = 1.0$) and not sufficiently enforcing distribution alignment ($\lambda_{\text{align}} = 0.01$).

The role of VAT (similarly to that of conditional entropy loss) is orthogonal to alignment objectives. Thus, we provide additional evaluations of combining support alignment and VAT (the "ASA-sq + VAT" entry in Table D.3) with alignment weight $\lambda_{\text{align}} = 1.0$. The good performance of such combination shows that:

- One could improve our current support alignment performance by using auxiliary objectives.
- Support alignment based method performs qualitatively different from distribution alignment based method, since the performance holds with stricter support alignment while distribution alignment needs to loosen the constraints considerably to reduce the performance degradation in the imbalanced setting.

Table D.3: Results of comparison of ASA with DANN and VADA across different values of the alignment loss weight $\lambda_{\text{align}}$ on STL→CIFAR data. Same setup and reporting metrics as Table 1.

| Algorithm | $\lambda_{\text{align}}$ | $\alpha = 0.0$ | | $\alpha = 1.0$ | | $\alpha = 1.5$ | | $\alpha = 2.0$ | |
|---|---|---|---|---|---|---|---|---|---|
| | | average | min | average | min | average | min | average | min |
| DANN | 0.01 | $72.3^{72.7}_{72.2}$ | $49.5^{50.8}_{48.8}$ | $70.6^{71.2}_{69.7}$ | $48.9^{51.2}_{41.5}$ | $68.5^{68.7}_{67.2}$ | $46.1^{50.0}_{36.2}$ | $65.9^{66.0}_{64.1}$ | $36.7^{39.4}_{29.9}$ |
| DANN | 0.1 | $75.3^{75.4}_{74.9}$ | $54.6^{56.6}_{54.2}$ | $69.9^{70.1}_{68.6}$ | $44.8^{45.1}_{40.7}$ | $64.9^{67.1}_{63.7}$ | $34.9^{36.8}_{33.9}$ | $63.3^{64.8}_{57.4}$ | $27.0^{28.5}_{21.2}$ |
| DANN | 1.0 | $77.2^{77.3}_{76.8}$ | $58.5^{59.4}_{56.7}$ | $66.3^{66.8}_{64.5}$ | $37.9^{41.6}_{37.5}$ | $62.8^{63.3}_{56.1}$ | $27.5^{28.9}_{24.6}$ | $58.7^{59.7}_{52.3}$ | $18.5^{20.5}_{17.2}$ |
| VADA | 0.01 | $74.4^{74.4}_{74.2}$ | $54.2^{55.4}_{52.6}$ | $71.7^{71.7}_{71.7}$ | $51.6^{52.0}_{45.0}$ | $69.5^{69.7}_{68.4}$ | $47.5^{49.8}_{40.0}$ | $65.9^{66.1}_{64.8}$ | $37.2^{39.4}_{35.3}$ |
| VADA | 0.1 | $76.7^{76.7}_{76.6}$ | $56.9^{58.3}_{53.5}$ | $70.6^{71.0}_{70.0}$ | $47.7^{48.8}_{44.0}$ | $66.1^{66.5}_{65.4}$ | $35.7^{39.3}_{33.3}$ | $63.2^{64.7}_{60.2}$ | $25.5^{28.0}_{25.2}$ |
| ASA-sq | 0.1 | $71.7^{71.9}_{71.7}$ | $52.9^{53.4}_{46.7}$ | $70.7^{71.0}_{70.4}$ | $51.6^{52.7}_{46.8}$ | $69.2^{69.3}_{69.2}$ | $45.6^{52.0}_{43.3}$ | $68.1^{68.2}_{67.2}$ | $44.7^{45.9}_{39.8}$ |
| ASA-sq + VAT | 1.0 | $74.2^{74.5}_{74.0}$ | $52.2^{52.5}_{51.9}$ | $72.2^{72.2}_{71.9}$ | $53.5^{53.6}_{45.4}$ | $70.6^{70.8}_{70.4}$ | $48.9^{52.3}_{45.6}$ | $67.4^{67.7}_{66.8}$ | $43.0^{46.0}_{39.4}$ |

## D.4 VISDA-17 EXPERIMENT SPECIFICATIONS

We use train and validation sets of the VisDA-17 challenge (Peng et al., 2017).

For the feature extractor we use ResNet-50 He et al. (2016) architecture with modified output size of the final linear layer. The feature representation is 256-dimensional vector. We use the weights from pre-trained ResNet-50 model (torchvision model hub) for all layers except the final linear layer. The classifier consists of a single linear layer. The discriminator is implemented by a 3-layer MLP with 1024 hidden units and leaky-ReLU activation.

We train all methods for 50 000 steps with batch size 36. We train the feature extractor, the classifier, and the discriminator with SGD. For the feature extractor we use learning rate 0.001, momentum 0.9, weight decay 0.001. For the classifier we use learning rate 0.01, momentum 0.9, weight decay 0.001. For the discriminator we use learning rate 0.005, momentum 0.9, weight decay 0.001. We perform a single discriminator update per 1 update of the feature extractor and the classifier. We linearly anneal the feature extractor's and classifier's learning rate throughout the training (50 000) steps. By the end of the training the learning rates of the feature extractor and the classifier are decreased by a factor of 0.05.

The weight for the classification term is constant and set to $\lambda_{\text{cls}} = 1$. We introduce schedule for the alignment weight $\lambda_{\text{align}}$. For all alignment methods we linearly increase $\lambda_{\text{align}}$ from 0 to 0.1 during the first 10000 steps. For all methods we use auxiliary conditional entropy loss on target examples with the weight $\lambda_{\text{ent}} = 0.1$.

For ASA we use history buffers of size 1000.

## D.5 HISTORY SIZE EFFECT AND EVALUATION OF SUPPORT DISTANCE

**History size effect**. To quantify the effects of mini-batch training mentioned in Section 3, we explore different sizes of history buffers on USPS→MNIST task with the label distribution shift $\alpha = 1.5$. The results are presented in Figure D.1 and Table D.4. Figure D.2 shows the distributions of outputs of the learned discriminator at the end of the training. While without any alignment objectives neither the densities nor the supports of $g^\psi_\sharp p^\theta_Z$ and $g^\psi_\sharp q^\theta_Z$ are aligned, both alignment methods approximately satisfy their respective alignment constraints. Compared with DANN results, ASA with small history size performs similarly to distribution alignment, while all history sizes are enough for support alignment. We also observe the correlation between distribution distance and target accuracy: under label distribution shifts, the better distribution alignment is achieved, the more target accuracy suffers. Note that with too big history buffers (e.g. $n = 5000$), we observe a sudden drop in performance and increases in distances. We hypothesize that this could be caused by the fact that the history buffer stores discriminator output values from the past steps while the discriminator parameters constantly evolve during training. As a result, for a large history buffer, the older items might no longer accurately represent the current pushforward distribution as they become outdated.

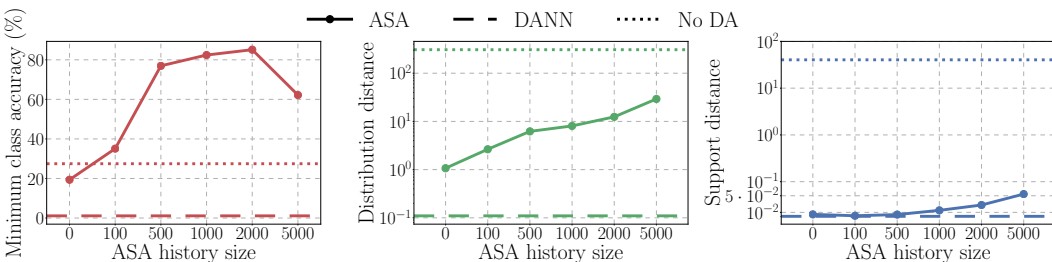

Figure D.1: Evaluation of history size effect for ASA on MNIST→USPS with the label distribution shift ($\alpha = 1.5$). The panels show (left to right): minimum class accuracy on target test set; Wasserstein distance $\mathcal{D}_W(g^\psi_\sharp p^\theta_Z, g^\psi_\sharp q^\theta_Z)$ between the pushforward distributions of source and target representations induced by the discriminator; SSD divergence $\mathcal{D}_\triangle(g^\psi_\sharp p^\theta_Z, g^\psi_\sharp q^\theta_Z)$ between the pushforward distributions. In each panel the dashed lines show the respective quantities for "No DA" and DANN methods.

**Direct evaluation of support distance.** In order to directly evaluate the ability of ASA (with history buffers) to enforce support alignment, we consider the setting of the illustrative experiment described in Section 5 (3-class USPS→MNIST adaptation with 2D feature extractor, $\alpha = 1.5$). We compare methods No DA, DANN, and ASA-abs (with different history buffer sizes). For each method we consider the embedding space of the learned feature extractor at the end of training and compute Wasserstein distance $\mathcal{D}_W(p_Z^\theta, q_Z^\theta)$ and SSD divergence $\mathcal{D}_\triangle(p_Z^\theta, q_Z^\theta)$ between the embeddings of source and target domain (note that we compute the distances in the original embedding space directly without projecting data to 1D with the discriminator). To ensure meaningful comparison of the distances between different embedding spaces, we apply a global affine transformation for each embedding space: we center the embeddings so that their average is 0 and re-scale them so that their average norm is 1. The results of this evaluation are shown in Table D.5. We observe that, compared to no alignment and distribution alignment (DANN) methods, ASA aligns the supports without necessarily aligning the distributions (in this imbalanced setting, distribution alignment implies low adaptation accuracy).

Table D.4: Analysis of effect history size parameter for ASA on USPS→MNIST with class label distribution shift corresponding to $\alpha = 1.5$. We report distribution and support distances between the pushforward distributions $g^\psi_\sharp p_Z^\theta$ and $g^\psi_\sharp q_Z^\theta$, as well as the value of discriminator's log-loss.

| Method | History size | Target accuracy (%) | | Distribution distances | | Log-loss |
|---|---|---|---|---|---|---|
| | | average | min | $\mathcal{D}_W(g^\psi_\sharp p_Z^\theta, g^\psi_\sharp q_Z^\theta)$ | $\mathcal{D}_\triangle(g^\psi_\sharp p_Z^\theta, g^\psi_\sharp q_Z^\theta)$ | |
| No DA | — | $71.28\,^{72.51}_{71.25}$ | $27.46\,^{37.26}_{24.21}$ | $307.56\,^{322.00}_{277.33}$ | $40.35\,^{46.06}_{32.10}$ | $00.05\,^{00.07}_{00.04}$ |
| DANN | — | $69.96\,^{71.25}_{63.89}$ | $01.11\,^{01.53}_{00.99}$ | $00.11\,^{00.11}_{00.10}$ | $00.00\,^{00.00}_{00.00}$ | $00.65\,^{00.65}_{00.65}$ |
| ASA-abs | 0 | $62.75\,^{64.35}_{61.78}$ | $19.36\,^{23.63}_{17.90}$ | $01.07\,^{01.15}_{00.99}$ | $00.01\,^{00.01}_{00.00}$ | $00.57\,^{00.58}_{00.56}$ |
| ASA-abs | 100 | $80.58\,^{81.73}_{78.22}$ | $35.09\,^{44.37}_{32.10}$ | $02.64\,^{02.70}_{02.15}$ | $00.00\,^{00.00}_{00.00}$ | $00.53\,^{00.53}_{00.52}$ |
| ASA-abs | 500 | $92.02\,^{92.76}_{90.56}$ | $76.96\,^{83.72}_{70.94}$ | $06.21\,^{06.48}_{05.69}$ | $00.00\,^{00.01}_{00.00}$ | $00.45\,^{00.45}_{00.45}$ |
| ASA-abs | 1000 | $92.54\,^{92.93}_{90.90}$ | $82.41\,^{85.43}_{74.53}$ | $08.06\,^{08.19}_{07.97}$ | $00.01\,^{00.02}_{00.01}$ | $00.41\,^{00.41}_{00.40}$ |
| ASA-abs | 5000 | $86.03\,^{87.50}_{84.86}$ | $62.19\,^{71.62}_{46.98}$ | $29.23\,^{29.63}_{24.54}$ | $00.05\,^{00.08}_{00.05}$ | $00.29\,^{00.30}_{00.29}$ |

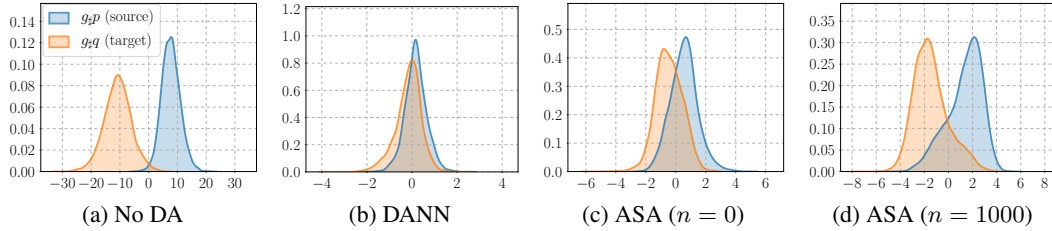

(a) No DA      (b) DANN      (c) ASA ($n = 0$)      (d) ASA ($n = 1000$)

Figure D.2: Kernel density estimates (in the discriminator output space) of $g^\psi_\sharp p_Z^\theta$, $g^\psi_\sharp q_Z^\theta$ at the end of the training on USPS→MNIST task with $\alpha = 1.5$. $n$ is the size of ASA history buffers.

Table D.5: Results of No DA, DANN, and ASA-abs (with different history sizes) on 3-class USPS→MNIST adaptation with 2D feature extractor and label distribution shift corresponding to $\alpha = 1.5$. We report average and minimum target class accuracy, as well as Wasserstein distance $\mathcal{D}_W$ and support divergence $\mathcal{D}_\triangle$ between source $p_Z^\theta$ and target $q_Z^\theta$ 2D embedding distributions. We report median (the main number), and 25 (subscript) and 75 (superscript) percentiles across 5 runs.

| Algorithm | History size | Accuracy (avg) | Accuracy (min) | $\mathcal{D}_W(p_Z^\theta, q_Z^\theta)$ | $\mathcal{D}_\triangle(p_Z^\theta, q_Z^\theta)$ |
|---|---|---|---|---|---|
| No DA | — | $63.0\,^{69.6}_{62.3}$ | $45.3\,^{53.6}_{37.9}$ | $0.78\,^{00.84}_{00.75}$ | $0.10\,^{0.10}_{0.10}$ |
| DANN | — | $75.6\,^{83.7}_{72.4}$ | $54.8\,^{55.1}_{49.6}$ | $0.07\,^{0.08}_{0.06}$ | $0.02\,^{0.02}_{0.02}$ |
| ASA-abs | 0 | $73.9\,^{84.1}_{73.4}$ | $61.8\,^{72.4}_{54.6}$ | $0.23\,^{0.47}_{0.22}$ | $0.03\,^{0.03}_{0.03}$ |
| ASA-abs | 100 | $88.5\,^{95.1}_{86.8}$ | $71.4\,^{93.3}_{70.6}$ | $0.54\,^{0.36}_{0.56}$ | $0.03\,^{0.03}_{0.03}$ |
| ASA-abs | 500 | $94.5\,^{94.7}_{88.7}$ | $89.0\,^{90.3}_{83.1}$ | $0.59\,^{0.64}_{0.55}$ | $0.03\,^{0.03}_{0.03}$ |
| ASA-abs | 1000 | $91.1\,^{93.0}_{91.1}$ | $85.6\,^{86.2}_{80.7}$ | $0.59\,^{0.62}_{0.55}$ | $0.03\,^{0.03}_{0.03}$ |
| ASA-abs | 2000 | $94.0\,^{94.7}_{91.2}$ | $88.6\,^{89.4}_{80.2}$ | $0.62\,^{0.66}_{0.58}$ | $0.03\,^{0.03}_{0.03}$ |
| ASA-abs | 5000 | $82.1\,^{83.9}_{81.8}$ | $68.9\,^{70.9}_{65.5}$ | $0.64\,^{0.67}_{0.63}$ | $0.04\,^{0.04}_{0.04}$ |

