# OpenReview forum: "Adversarial Support Alignment"
_ICLR.cc/2022/Conference — ICLR 2022 Spotlight_

### Official Review · Reviewer_FsUp · 2021-10-29

**Correctness:** 4
**Technical Novelty And Significance:** 3
**Empirical Novelty And Significance:** 3
**Recommendation:** 8
**Confidence:** 4

**Main Review:**

# Strengths
* Rreducing divergence measures between multi-dimensional distributions to unidimensional ones, whether it concerns the support alignment (Theorem 2.1) or the Wasserstein type divergences (Proposition 4.3).
* The interest in the minimum accuracy per class, which is a metric that is rarely considered in the domain adaptation literature to the best of my knowledge, but that is important in real-world applications.

# Weaknesses

* There is no conclusion summarizing the main findings of the paper at least.

* Experimental part:

    * It is hard to tell how much of the success can be attributed to the developments in the theory, especially that apart from VADA, other methods do not use a conditional entropy loss in their original implementations, whereas this latter loss is added with a coefficient $\lambda_{\text{ent}} = 0.1$. Also, the expression of this unsupervised loss should be mentioned in the supplementary material (or at least a reference to its expression, e.g. from [1]). I am not advocating for the idea that an algorithm should surpass all of the state-of-the-art methods, but it is instructive for the community to see if the proposed theory offers the same order of performance.
    * The domain discriminator has a single update per 1 update of feature extractor and classifier. This does not go in line with the theoretical findings that only concern the optimal discriminator $f^*(x)$, i.e. after convergence. Such a detail needs to be mentioned in the main paper.

# Other comments
* In Proposition 2.3, a counter-example is provided concerning the equivalence between aligning the supports in the original spaces and their 1D push-forward counterparts induced by a domain discriminator. While the establishment of the result concerning the JS divergence discriminator relies on densities, the provided counter-example concerns discrete distributions. It would be better to provide a counter-example with distributions having densities.
* Appendix A: it should be $f$ rather than $f_W^*$ in the operand of the $\sup_{f:Lip(f)\leq 1}$ operator.
## Typos:
* Before Remark 2.1.1: "techincal" --> "technical"
* Paragraph **Setup**: "optimizied" --> "optimized"

# Questions
* In Proposition B.1, after slicing the distributions by $\theta \in \mathbb S^1$, are the supports of the push-forward distributions equal? Because $f^\theta \\#q$ will have no supports on a segment of length $2$, centered at zero.
* I am not sure if I understand the notation $q^Y(y) \propto \sigma(y)^{-\alpha}$. Does it imply that for example, we can have $q^Y(1) \propto K^{-\alpha}$ ?
* Provided that I understood correctly the previous point, when $\alpha$ is very large, it means that some class is almost absent, leading to the partial DA setting. Did you test the approach in this regime?
* Did you notice any acceleration of the algorithm via using the strongly convex squared distance rather than the absolute one ?
* When you mention "all alignment methods", do you refer to only **ASA-sq** and **ASA-abs**, or to all of the other approaches as well? If the latter case holds, why not use the default parameters of the methods (e.g. $\lambda_{\text{align}} = 0.01, 1$ respectively for VADA and DANN, instead of the fixed value 0.1) ?
* Is it possible to establish theoretical guarantees on the convergence of the empirical support divergence to its true counterpart? and can this aspect be related to the observation that large buffer sizes imply a drop in performance?

# References
[1] Shu, R., Bui, H. H., Narui, H., & Ermon, S. (2018). A dirt-t approach to unsupervised domain adaptation. *arXiv preprint arXiv:1802.08735*.

**Summary Of The Paper:**

The authors propose a weak alignment of distributions consisting in equalizing their supports. They justify this choice by the recent advancements in Domain Adaptation that show the drawbacks of a strict alignment of marginal distributions. A distance between supports is defined as a modification of the Hausdorff distance between sets. The support divergence is further proven to be a limit case of a previously proposed weak Wasserstein distance requiring only the boundedness of the density ratio, thus defining a continuum of relaxations of divergences.  For all of the divergences in this continuum, a theoretical result shows the possibility of reducing the original problem of computing the divergence to unidimensional distributions, obtained via a domain discriminator minimizing a Jensen-Shannon divergence. The theoretical findings are used to propose an algorithm that is tested in the class imbalanced domain adaptation setting, highlighting the interest of the approach.

**Summary Of The Review:**

The theory established in the paper is interesting for the community, especially with the general result of reducing the comparison between distributions to a uni-dimensional one. However, there are some gaps between the theory and the proposed algorithm, especially concerning the use of the unsupervised conditional entropy loss on the target domain.

# Post-rebuttal
As a result of the authors's detailed response, I update my review score to 8.

---

> ### Author Response · Authors · 2021-11-16
> **Response to reviewer FsUp (Part 1)**
>
> We thank the reviewer for the thorough review of our paper and the thoughtful feedback. We provide an overview of our response to the major points first. The specific details about all questions raised in the review are provided in subsequent parts of the response.
>
> * Regarding the usage of conditional entropy loss in our experiment (details in Parts 2 & 3):
>
>   We provide the results of an ablation study comparing the performance of all domain adaptation methods with/without the conditional entropy objective (details in Part 2 & 3). The results demonstrate that using the conditional entropy improves all methods uniformly and its benefits are orthogonal to the benefits provided by support alignment.
>
> * Regarding single discriminator update (details in Part 2):
>
>   We will include the discussion of this detail in the revised version of the paper.
>
>   Using single update of the discriminator per 1 update of the feature extractor is the common practice in the implementation of existing adversarial alignment methods (distribution alignment and relaxed distribution alignment), which also assume optimality of the discriminator in their theory. The reason for this choice is the computational cost: several steps of discriminator per one step of feature extractor in adversarial training takes significantly longer time.
>
> We are happy to answer any further questions regarding the paper.

---

> > ### Author Response · Authors · 2021-11-16
> > **Response to reviewer FsUp (Part 7)**
> >
> > | Algorithm   | $\lambda_{\text{align}}$   | $\alpha = 0.0$                                    | $\alpha = 1.0$                                | $\alpha = 1.5$                                         | $\alpha = 2.0$                                    |
> > |:------------|:-----------------------|:--------------------------------------------------|:--------------------------------------------------|:--------------------------------------------------|:--------------------------------------------------|
> > | DANN        | $0.1$                     | $ 75.3_{74.9}^{75.4} $ $ 54.6_{54.2}^{56.6} $ | $ 69.9_{68.6}^{70.1} $ $ 44.8_{40.7}^{45.1} $ | $ 64.9_{63.7}^{67.1} $ $ 34.9_{33.9}^{36.8} $ | $ 63.3_{~57.4}^{64.8} $ $ 27.0_{21.2}^{28.5} $ |
> > | DANN        | $1.0$                     | $ 77.2_{76.8}^{77.3} $ $ 58.5_{56.7}^{59.4} $ | $ 66.3_{64.5}^{66.8} $ $ 37.9_{37.5}^{41.6} $ | $ 62.8_{56.1}^{63.3} $ $ 27.5_{24.6}^{28.9} $ | $ 58.7_{~52.3}^{59.7} $ $ 18.5_{17.2}^{20.5} $ |
> > | VADA        | $0.1$                     | $ 76.7_{76.6}^{76.7} $ $ 56.9_{53.5}^{58.3} $ | $ 70.6_{70.0}^{71.0} $ $ 47.7_{44.0}^{48.8} $ | $ 66.1_{65.4}^{66.5} $ $ 35.7_{33.3}^{39.3} $ | $ 63.2_{~60.2}^{64.7} $ $ 25.5_{25.2}^{28.0} $ |
> > | VADA        | $0.01$                    | $ 74.4_{74.2}^{74.4} $ $ 54.2_{52.6}^{55.4} $ | $ 71.7_{71.7}^{71.7} $ $ 51.6_{45.0}^{52.0} $ | $ 69.5_{68.4}^{69.7} $ $ 47.5_{40.0}^{49.8} $ | $ 65.9_{~64.8}^{66.1} $ $ 37.2_{35.3}^{39.4} $ |
> > | ASA-sq       | $0.1$                     | $ 71.7_{71.7}^{71.9} $ $ 52.9_{46.7}^{53.4} $ | $ 70.7_{70.4}^{71.0} $ $ 51.6_{46.8}^{52.7} $ | $ 69.2_{69.2}^{69.3} $ $ 45.6_{43.3}^{52.0} $ | $ 68.1_{~67.2}^{68.2} $ $ 44.7_{39.8}^{45.9} $ |
> > | ASA-sq + VAT | $1.0$                     | $ 74.2_{74.0}^{74.5} $ $ 52.2_{51.9}^{52.5} $ | $ 72.2_{71.9}^{72.2} $ $ 53.5_{45.4}^{53.6} $ | $ 70.6_{70.4}^{70.8} $ $ 48.9_{45.6}^{52.3} $ | $ 67.4_{66.8}^{67.7} $ $ 43.0_{39.4}^{46.0} $ |

---

> > ### Author Response · Authors · 2021-11-17
> > **Response to reviewer FsUp (Part 6)**
> >
> > > Did you notice any acceleration of the algorithm via using the strongly convex squared distance rather than the absolute one?
> >
> > We consider two choices of distance since the versions of SSD divergence with absolute distance and squared distance provide different training signals in the 1D discriminator output space. The gradient of the absolute distance is $\pm 1$ and it does not depend on the distance between two points while for the square distance the magnitude of the gradient is proportional to the distance.
> >
> > In our experiments, we did not observe an acceleration of the algorithm with the square distance. While strong convexity is a useful property of the squared distance, there are no guarantees that this benefit will lead to better training dynamics in experiments in the 3-player game setting with the models represented by deep neural networks (the training dynamics is defined by co-evolution of the players and the objective is not convex w.r.t. to the networks’ parameters).
> >
> > ---
> >
> > > When you mention "all alignment methods", do you refer to only **ASA-sq** and **ASA-abs**, or to all of the other approaches as well? If the latter case holds, why not use the default parameters of the methods (e.g. $\lambda_{\text{align}} = 0.01, 0.1$ respectively for VADA and DANN, instead of the fixed value 0.1) ?
> >
> > All alignment methods refer to all approaches except for source-only training/no DA. The alignment weight describes how much the algorithm focuses on its respective alignment objective. The higher the value is, the stricter the constraint is enforced. We used a fixed value of 0.1 for alignment weight across because from our observation this value is not too high that all distribution alignment based methods suffer strong performance drop under label distribution shift, and it is not too low that the point of this empirical comparison, characterizing how different types of alignment constraints affect the real-world performance, is lost.
> >
> > Per your suggestions, we provide the additional results (see table in Part 7) for DANN with weight 1 and VADA with weight 0.01 on STL->CIFAR task. Note that DANN with a higher alignment weight performs better in the balanced ($\alpha=0$) setting and worse in the imbalanced ($\alpha>0$) setting compared to a lower weight 0.1, as the distribution alignment constraint is enforced stricter. VADA is essentially distribution alignment + VAT (virtual adversarial training), and we see the same trend: with lower alignment weight (0.01), it performs worse in the balanced setting and better in the imbalanced setting compared to a higher weight (0.1). Weight 0.1 is a middle ground between having poor performance in the imbalanced setting (weight 1) and not really enforcing distribution alignment (weight 0.01).
> >
> > As a side note, the role of VAT (similarly to that of conditional entropy loss) is orthogonal to alignment objectives. Thus, we provide additional evaluations of combining support alignment and VAT (the “ASA-sq + VAT” entry in the table) with alignment weight 1. The good performance of such combination shows that (a) one could improve our current support alignment performance by using auxiliary objectives, (b) support alignment based method performs qualitatively different from distribution alignment based method, since the performance holds with stricter support alignment while distribution alignment needs to loosen the constraints considerably to reduce the performance degradation in the imbalanced setting.
> >
> > ---
> >
> > > Is it possible to establish theoretical guarantees on the convergence of the empirical support divergence to its true counterpart? and can this aspect be related to the observation that large buffer sizes imply a drop in performance?
> >
> > As we discussed in the paper the support divergence can be viewed through the lens of optimal transport. While we did not perform the convergence analysis, we suspect that it might be possible to adapt the existing analysis of the convergence of the empirical estimators of Wasserstein distance (e.g. [1]).
> >
> > We hypothesize that the drop in performance for a large history buffer size can be caused by the fact that the history buffer stores discriminator output values from the past steps while the discriminator parameters constantly evolve during training. As a result, for a large history buffer, the older items might no longer accurately represent the current pushforward distribution as they become outdated.
> >
> > [1] Weed, J., & Bach, F. (2019). Sharp asymptotic and finite-sample rates of convergence of empirical measures in Wasserstein distance. Bernoulli, 25(4A), 2620-2648.

---

> > ### Author Response · Authors · 2021-11-17
> > **Response to reviewer FsUp (Part 5)**
> >
> > > In Proposition B.1, after slicing the distributions by $\theta \in \mathbb{S}^1$, are the supports of the pushforward distributions equal? Because $f^\theta\sharp q$ will have no supports on a segment of length , centered at zero.
> >
> > The supports of the push-forward distributions are equal, since $f^\theta_\sharp q$ in fact has support on segment [-2, 2]. One way to show this is the following.
> >
> > Informally, the reason is that the linear mapping $f^\theta(x) =  \langle \theta, x \rangle$  projects the points onto a line (e.g, for $\theta = (0, 1)$ the line is the horizontal  axis, $f^\theta((x_1, x_2)) = x_1$) and projection of the support of $q$ is the segment $[-2, 2]$ for any $\theta$.
> >
> > ### Slightly more detailed argument
> >
> > Let $x = (x_1, x_2)$ denote a 2D point with Cartesian coordinates $x_1, x_2$. For a fixed 2D direction $\theta = (\theta_1, \theta_2): \|\theta\| = 1$ the linear mapping $f^\theta$ is defined as $f^\theta(x) = \langle \theta, x \rangle = \theta_1 x_1 + \theta_2 x_2$. For a real number $a$ and $\varepsilon > 0$ consider the probability of the interval $(a - \varepsilon, a + \varepsilon)$ under the pushforward distribution $f^{\theta}_\sharp q$. This probability is equal to the probability of the random variable $f^{\theta}(x)$ taking value in the interval $(a - \varepsilon, a + \varepsilon)$ for $x \sim q$, or formally
> >
> > $\mathbb{P}_{t \sim f^{\theta}_\sharp q}(t \in (a - \varepsilon, a + \varepsilon))$
> >
> > $= \mathbb{P}_{x \sim q}(f^\theta(x) \in (a - \varepsilon, a + \varepsilon))$.
> >
> > Let us choose $\theta = (1, 0)$ (the result is the same for any direction $\theta$ due to the rotational symmetry). For $\theta = (1, 0)$, $f^\theta((x_1, x_2)) = x_1$ and the probability we are interested in is $\mathbb{P}_{x \sim q}(x_1 \in (a - \varepsilon, a + \varepsilon))$
> > $= \iint I[x_1 \in (a - \varepsilon, a + \varepsilon)] q(x_1, x_2) dx_1 dx_2$ where $I[\cdot]$ denotes the indicator function. Note that under distribution $q$ we have non-zero probability of sampling point $x = (x_1, x_2)$ lying in a vertical stripe $a - \varepsilon < x_1 < a + \varepsilon$ for any offset $a$ of the stripe in $[-2, 2]$ (including $a \in [-1, 1]$).
> >
> > ---
> >
> > > I am not sure if I understand the notation $q^Y(y) \varpropto \sigma(y)^{-\alpha}$. Does it imply that for example, we can have  $q^Y(y) \varpropto K^{-\alpha}$?
> >
> > Yes, your understanding of the notation $q^Y(y) \varpropto \sigma(y)^{-\alpha}$ is correct. In more details, consider an example of a permutation of classes $\sigma$: $\{\sigma(y)\}_{y=1}^K = \{K, K-1, K-2, \ldots, 2, 1\}$. Then
> >
> > $q^Y(1) \varpropto \sigma(1)^{-\alpha} \varpropto K^{-\alpha}$
> >
> >  and after normalization the exact value of the probability is $q^Y(1) = \frac{K^{-\alpha}}{\sum_{i=1}^K i^{-\alpha}}$.
> >
> > ---
> >
> > > Provided that I understood correctly the previous point, when $\alpha$ is very large, it means that some class is almost absent, leading to the partial DA setting. Did you test the approach in this regime?
> >
> > We did not explore the partial domain adaptation setting. In all our experiment setups, we made sure that there is a reasonable number of samples presented for each class (so that the evaluation is informative), since the minimum class accuracy is an important metric in our evaluation. Nevertheless, our approach can naturally extend to the partial DA setting, and we provide a discussion below.
> >
> > In the partial DA setting where some classes of the source domain do not appear in the target domain, instead of our proposed support alignment approach, a “support containment” method is more suitable, i.e. the support of target representation distribution should be contained in the support of source representation distribution. This can be easily done in our framework, since SSD divergence is essentially the sum of two support containment losses. More specifically, following our notation of having $p$ as source domain distribution and $q$ as target domain distribution (we omit the feature extractor part for simplicity), we could consider only the second term of equation (1):
> >
> > $\mathbb{E}_q [d(x,\operatorname{supp}(p))]$,
> >
> > and update the representations of $q$ in practice by minimizing the corresponding term in equation (5): $\sum\limits_{j=1}^{n+m}d(v_j^q, v^p)$.
> >
> > While in this paper, we primarily focus on support alignment as a natural constraint for domain adaptation with the same set of classes across the domains, in the revised version of the paper, we will include the discussion of potential extensions such as support containment for partial domain adaptation. We believe that our methodology opens possibilities for the design of more nuanced and structured alignment constraints.

---

> > ### Author Response · Authors · 2021-11-17
> > **Response to reviewer FsUp (Part 4)**
> >
> > > In Proposition 2.3, a counter-example is provided concerning the equivalence between aligning the supports in the original spaces and their 1D push-forward counterparts induced by a domain discriminator. While the establishment of the result concerning the JS divergence discriminator relies on densities, the provided counter-example concerns discrete distributions. It would be better to provide a counter-example with distributions having densities.
> >
> > Indeed, for Proposition 2.3 one can construct counterexample distributions with densities. Note that our proof only relies on symmetry and the fact that the dual Wasserstein discriminator focuses on the differences between densities, whereas the log-loss discriminator focuses on the ratios between densities. Below we provide detailed proof which extends our original counterexample to distributions with densities. The idea is to replace each of the points $\{-1, 0, 1\}$ with a short segment at the respective location and define uniform distributions within each segment.
> >
> > We will update this example in the revised version of the paper.
> >
> > ### Updated proof
> >
> > Consider a 1-dimensional Euclidean space $\mathbb{R}$. Let $\operatorname{supp}(p) = [-\frac{1}{2}, \frac{1}{2}]\cup[1,2]$ with $p([-\frac{1}{2}, \frac{1}{2}]) = \frac{3}{4}$ and $p([1, 2])=\frac{1}{4}$. Let $\operatorname{supp}(q) = [-2, -1]\cup[-\frac{1}{2}, \frac{1}{2}]\cup[1, 2]$ with $q([-2, -1])=\frac{1}{4}, q([-\frac{1}{2}, \frac{1}{2}]) = \frac{1}{4}$ and $q([1, 2])=\frac{1}{2}$. The supports of $p$ and $q$ consist of disjoint closed intervals, and we assume uniform distribution within each of these intervals, i.e. $p$ has density $p(x) = \frac{3}{4}, \forall x \in [-\frac{1}{2}, \frac{1}{2}]; p(x) = \frac{1}{4}, \forall x \in[1, 2]$ and $q$ has densitiy $q(x) = \frac{1}{4}, \forall x \in [-2, -1]; q(x) = \frac{1}{4}, \forall x \in [-\frac{1}{2}, \frac{1}{2}]; q(x) = \frac{1}{2}, \forall x\in [1,2]$. Clearly, $\operatorname{supp}(p)\neq\operatorname{supp}(q)$.
> >
> > The optimal dual Wasserstein discriminator is the solution of $\sup\limits_{f:\operatorname{Lip}(f)\leq1}\int_{-\frac{1}{2}}^{\frac{1}{2}}\frac{3}{4}f(x)dx+\int_{1}^{2}\frac{1}{4}f(x)dx-\int_{-2}^{-1}\frac{1}{4}f(x)dx-\int_{-\frac{1}{2}}^{\frac{1}{2}}\frac{1}{4}f(x)dx-\int_{1}^{2}\frac{1}{2}f(x)dx$, which simplifies to $\sup\limits_{f:\operatorname{Lip}(f)\leq1}-\frac{1}{4}\int_{-2}^{-1}f(x)dx+\frac{1}{2}\int_{-\frac{1}{2}}^{\frac{1}{2}}f(x)dx-\frac{1}{4}\int_{1}^{2}f(x)dx$. Since the cost function and the constraints are invariant to replacing the function $f(x)$ with its symmetric reflection ($g(x) = f(-x))$, there exists a symmetric maximizer $f_W^*(x) = f_W^*(-x), \forall 1\leq x\leq2$, making $\operatorname{supp}({f_W^*}_\sharp p)=\operatorname{supp}({f_W^*}_\sharp q)$.
> >
> >
> > Thank you for pointing out the typos. We will fix them in the updated version of the paper.

---

> > ### Author Response · Authors · 2021-11-17
> > **Response to reviewer FsUp (Part 3)**
> >
> > | Algorithm   | $\lambda_{\text{ent}}$ | $\alpha = 0.0$                                    | $\alpha = 1.0$                                | $\alpha = 1.5$                                         | $\alpha = 2.0$                                    |
> > |:------------|:-----------------------|:--------------------------------------------------|:--------------------------------------------------|:--------------------------------------------------|:--------------------------------------------------|
> > | DANN        | $0$                    |$ 74.6_{74.1}^{75.1} $ $ 51.5_{49.9}^{55.0} $ | $ 68.4_{67.0}^{69.2} $ $ 43.2_{41.2}^{43.7} $ | $ 65.7_{62.8}^{65.9} $ $ 35.5_{29.6}^{36.2} $ | $ 62.5_{60.0}^{64.6} $ $ 27.5_{25.7}^{27.5} $ |
> > | IWDAN       | $0$                    | $ 70.4_{70.2}^{70.7} $ $ 47.2_{46.8}^{48.0} $ | $ 68.6_{68.4}^{68.8} $ $ 43.6_{43.2}^{46.3} $ | $ 66.7_{66.0}^{67.9} $ $ 44.7_{43.3}^{46.2} $ | $ 63.9_{62.9}^{66.1} $ $ 36.5_{32.7}^{37.3} $ |
> > | IWCDAN      | $0$                    | $ 70.1_{70.0}^{70.8} $ $ 50.5_{49.1}^{50.8} $ | $ 68.6_{68.2}^{69.4} $ $ 44.2_{41.2}^{45.8} $ | $ 66.0_{65.9}^{66.0} $ $ 45.0_{43.7}^{47.8} $ | $ 63.8_{62.3}^{64.1} $ $ 37.3_{33.6}^{37.7} $ |
> > | sDANN-4     | $0$                  | $ 69.4_{68.8}^{70.0} $ $ 46.5_{45.1}^{49.7} $ | $ 69.6_{69.3}^{69.7} $ $ 49.1_{47.4}^{49.2} $ | $ 68.0_{67.8}^{68.6} $ $ 48.2_{42.6}^{48.8} $ | $ 66.3_{64.2}^{66.4} $ $ 40.7_{36.6}^{42.9} $ |
> > | ASA-sq      | $0$                  | $ 69.9_{69.9}^{70.3} $ $ 48.0_{46.6}^{50.1} $ | $ 68.8_{68.6}^{68.9} $ $ 47.3_{45.3}^{49.3} $ | $ 68.1_{67.2}^{68.7} $ $ 45.4_{45.2}^{47.8} $ | $ 65.7_{65.6}^{66.4} $ $ 43.6_{41.3}^{45.0} $ |
> > | ASA-abs     | $0$                  | $ 69.8_{68.9}^{70.0} $ $ 45.7_{45.4}^{48.0} $ | $ 68.4_{68.4}^{68.6} $ $ 44.3_{44.0}^{46.8} $ | $ 67.9_{67.0}^{68.1} $ $ 46.6_{40.4}^{48.4} $ | $ 66.3_{65.7}^{66.9} $ $ 41.6_{40.3}^{44.9} $ |
> >
> > | Algorithm   | $\lambda_{\text{ent}}$ | $\alpha = 0.0$                                    | $\alpha = 1.0$                                | $\alpha = 1.5$                                         | $\alpha = 2.0$                                    |
> > |:------------|:-----------------------|:--------------------------------------------------|:--------------------------------------------------|:--------------------------------------------------|:--------------------------------------------------|
> > | DANN        | $0.1$                  | $ 75.3_{74.9}^{75.4} $ $ 54.6_{54.2}^{56.6} $ | $ 69.9_{68.6}^{70.1} $ $ 44.8_{40.7}^{45.1} $ | $ 64.9_{63.7}^{67.1} $ $ 34.9_{33.9}^{36.8} $ | $ 63.3_{57.4}^{64.8} $ $ 27.0_{21.2}^{28.5} $ |
> > | IWDAN       | $0.1$                  | $ 69.9_{69.9}^{70.7} $ $ 50.5_{47.9}^{50.6} $ | $ 68.7_{68.6}^{69.1} $ $ 45.8_{44.8}^{50.5} $ | $ 67.1_{65.9}^{67.3} $ $ 44.7_{40.4}^{44.8} $ | $ 64.4_{63.6}^{64.9} $ $ 36.8_{34.5}^{37.9} $ |
> > | IWCDAN      | $0.1$                  | $ 70.1_{70.1}^{70.2} $ $ 47.8_{42.4}^{49.3} $ | $ 69.4_{69.1}^{69.4} $ $ 47.1_{46.3}^{51.3} $ | $ 66.1_{65.0}^{67.2} $ $ 39.9_{37.7}^{40.8} $ | $ 64.5_{63.9}^{65.1} $ $ 37.0_{35.5}^{40.2} $ |
> > | sDANN-4     | $0.1$                | $ 71.8_{71.7}^{72.1} $ $ 52.1_{52.1}^{52.8} $ | $ 71.1_{70.4}^{71.7} $ $ 49.9_{48.1}^{51.8} $ | $ 69.4_{68.7}^{70.0} $ $ 48.6_{43.5}^{49.0} $ | $ 66.4_{66.2}^{67.9} $ $ 39.0_{33.6}^{47.1} $ |
> > | ASA-sq      | $0.1$                | $ 71.7_{71.7}^{71.9} $ $ 52.9_{46.7}^{53.4} $ | $ 70.7_{70.4}^{71.0} $ $ 51.6_{46.8}^{52.7} $ | $ 69.2_{69.2}^{69.3} $ $ 45.6_{43.3}^{52.0} $ | $ 68.1_{67.2}^{68.2} $ $ 44.7_{39.8}^{45.9} $ |
> > | ASA-abs     | $0.1$                | $ 71.6_{71.2}^{71.7} $ $ 49.0_{48.4}^{53.5} $ | $ 70.9_{70.8}^{71.0} $ $ 49.2_{47.3}^{50.0} $ | $ 69.6_{69.6}^{69.9} $ $ 43.2_{42.1}^{49.5} $ | $ 67.8_{66.6}^{68.2} $ $ 40.9_{35.4}^{49.0} $ |

---

> > ### Author Response · Authors · 2021-11-17
> > **Response to reviewer FsUp (Part 2)**
> >
> > > There is no conclusion summarizing the main findings of the paper at least.
> >
> > We will add a conclusion section in our updated version of the paper.
> >
> > ---
> >
> > > It is hard to tell how much of the success can be attributed to the developments in the theory, especially that apart from VADA, other methods do not use a conditional entropy loss in their original implementations, whereas this latter loss is added with a coefficient . Also, the expression of this unsupervised loss should be mentioned in the supplementary material (or at least a reference to its expression, e.g. from [1]). I am not advocating for the idea that an algorithm should surpass all of the state-of-the-art methods, but it is instructive for the community to see if the proposed theory offers the same order of performance.
> >
> > We use this auxiliary objective for all domain adaptation methods in order to ensure a fair comparison between the methods, since VADA uses this objective and generally this auxiliary objective improves the performance of all domain adaptation methods. The use of this loss is orthogonal to all alignment objectives, since it acts as an additional regularization of the embeddings of the unlabeled target examples by pushing the embeddings away from the classifier’s decision boundary.
> >
> > In order to quantify the improvements of the support alignment objective and the conditional entropy objective in separation, we conduct an ablation study. In addition to the results reported in the paper, we evaluate all domain adaptation methods (except VADA which uses the conditional entropy in the original implementation) on STL->CIFAR task without the conditional entropy loss ($\lambda_{\text{ent}} = 0$). The results of the ablation study are presented in the tables in Part 3. The evaluation metrics are the same as in Table 1 (in each cell the first set of numbers shows the statistics of the class-average target accuracy, and the second set of numbers shows the statistics of the worst target class accuracy).
> >
> > We observe that the effect of the auxiliary conditional entropy is essentially the same for all methods across all imbalance levels: with $\lambda_{\text{ent}}=0.1$ the accuracy either improves (especially the average class accuracy) or roughly stays on the same level. The relative ranking of distribution alignment, relaxed distribution alignment, and support alignment methods is the same with both $\lambda_{\text{ent}}=0.0$ and $\lambda_{\text{ent}}=0.1$. The results confirm that the benefits of support alignment approach and conditional entropy are orthogonal. We will include the results of the presented ablation study in the appendix.
> >
> > Per your suggestion, we will also include the expression for the conditional entropy objective in the updated version of the paper.
> >
> > We would also like to note that the support alignment constraint is the least restrictive of the alignment constraint. We believe that due to this property the support alignment can be more effectively combined with auxiliary regularizers (conditional cross entropy is one example). We leave further exploration of this direction for future work.
> >
> > ---
> >
> > > The domain discriminator has a single update per 1 update of feature extractor and classifier. This does not go in line with the theoretical findings that only concern the optimal discriminator, i.e. after convergence. Such a detail needs to be mentioned in the main paper.
> >
> > We will include a discussion of the suboptimality of the discriminator trained with a single update per feature extractor step.
> >
> > While not reaching the theoretical limit, using a single update of discriminator per one step of feature extractor is the common practice in the implementation of adversarial alignment methods. This includes all the distribution alignment and relaxed distribution alignment based methods, where the optimality of the discriminator is assumed in theory. The primary practical concern here is computation cost. More steps of the discriminator take substantially longer to train, but they still will not guarantee the optimality of the discriminator, which requires at least full training until convergence. Due to the sheer number of experiments (methods, tasks, imbalance levels, and seeds) we presented in the paper, we could not entertain many updates of discriminator per one step of feature extractor with our computation resources.

---

> ### Comment · Reviewer_FsUp · 2021-11-28
> **Response after rebuttal**
>
> I thank the reviewers for the effort they have put into assessing this paper.
> I also thank the authors for the considerably thorough response they provided for the different points I raised, which ultimately led to improve my understanding of the link between the developed theory and the observed theoretical performance.
> Considering that I am already convinced by the theoretical interest of the proposed approach to the community, I update my score to 8: good paper.

---

### Official Review · Reviewer_nH3D · 2021-11-02

**Correctness:** 3
**Technical Novelty And Significance:** 2
**Empirical Novelty And Significance:** 2
**Recommendation:** 3
**Confidence:** 3

**Main Review:**

+ Strengths:

It seems that the authors propose a "new" divergence to align supports of distributions, and empirically show its advantages in domain adaption.

+ Weakness:

The proposed SSD is closely related to the Chamfer divergence, if not the same (see ArXiv1612.00603).

+ Detailed comments:

1. Could the authors discuss the differences between the proposed SSD with the Chamfer divergence? (see ArXiv1612.00603, ArXiv2102.04014)?

2. It seems unclear about the motivation of using the 1d-projection? Is it better to consider the original space? What are the advantages of using the 1d-projection?

3. I also wonder about the advantages of using the 1d-projection by the optimal log-loss discriminator. Why is it favored over other projects? (is it suboptimal/optimal in some sense?)

4. The authors emphasize considering the alignment for the supports of distributions instead of comparing distributions. Are there any reasons to consider the expectation instead of average in equation (1) for SSD? [In case, only supports are important, I wonder why SSD  needs their "weights" in its definition?]

5. For experiments, I think the proposed method is closely related to optimal transport-based approaches for domain adaption. However, it seems that there are no such baselines yet (e.g., using standard OT, using sliced-Wasserstein, or their variants for domain adaption).



**Summary Of The Paper:**

The authors propose symmetric support difference (SSD) divergence to align supports of distributions. The authors further propose to use the optimal log-loss discriminator to project supports into 1-dimensional space for SSD. The authors also illustrate the advantages of the proposed methods for domain adaptation.

**Summary Of The Review:**

It seems that the proposed method (SSD) is closely related to Chamfer divergence if not the same (however, there are no discussions about their relationship yet). For experiments, in my opinion, the proposed approach is similar to optimal transport. Therefore, the authors should compare the proposed approach with other optimal transport-based approaches for domain adaption.

---

> ### Author Response · Authors · 2021-11-11
> **Response to Reviewer nH3D (Part 1)**
>
> Thank you for the comments and feedback. We provide our responses to your questions below.
>
> ### 1. Regarding our contribution as proposing SSD divergence and its relationship with Chamfer divergence:
>
> We provided an introduction to SSD divergence so as to motivate it specifically from the point of view of measuring distribution support differences. In Section 6, we discussed how SSD divergence relates to Relaxed Word Mover’s Distance (RWMD) which is identical to Chamfer divergence. We thank the reviewer for the additional reference and will add it alongside the discussion of RWMD. Note that SSD divergence is provided as a general divergence measure suitable for continuous densities as well, while both Chamfer divergence and RWMD are stated for discrete point sets. This distinction is necessary for our main theoretical analysis (Section 2.2 and 4.1), and also relates to reviewer BDME’s comments on “continuous” vs “discrete” support alignment.
>
> Please note that proposing SSD divergence (Section 2.1) is only one part of our contributions. Our paper introduces the concept and methodology of support alignment. In Section 2.2, we showed that log-loss discriminator preserves support differences in its one-dimensional output space. This enables us to do support alignment efficiently in the 1D discriminator output space. This property does not hold for all discriminators, including Wasserstein. On the basis of this analysis, our proposed algorithm (ASA) is then described in Section 3, and shown to be empirically effective in Section 5. In Section 4, we placed different notions of alignment – distribution alignment, relaxed distribution alignment, and support alignment – within a coherent spectrum from the point of view of optimal transport, characterizing their relationships, both theoretically in terms of their objectives and practically in terms of their algorithms.
>
>
> ### 2. Regarding the reason for using 1D projection:
>
> By the Cramér-Wold theorem, distributional differences are visible in appropriately selected 1D projections. Adjusting the representations based on the projected 1D signal is substantially easier than operating directly in the original space. Specifically, it would be challenging to measure SSD divergence in high dimensional spaces with a fair number of samples (this is true for our experimental setting for domain adaptation). Consider the following related analogy. One could calculate the Wasserstein metric (OT distance) between two sets of points in high dimensions, but it is often much easier to solve the problem in the dual, finding a discriminator that maps distributional differences onto 1D axis of function values. We are exploiting the same advantage by using a log-loss discriminator for support alignment. Moreover, support differences have to be repeatedly calculated. In the domain adaptation setting, embeddings are parametrically calculated from original examples. Any change in the parameters will change all the embeddings. Highlighting such evolving support differences is faster with warm started discriminators which reduce each round of support difference calculations to essentially 1D sorting.
>
>
> ### 3. Regarding the comparison between the optimal log-loss discriminator projection and other projections:
>
> Compared to distribution alignment our problem is more subtle. We would like support differences to manifest themselves as support differences in the 1D projection as well. We have shown that this is true with the log-loss discriminator (Theorem 2.1) but does not necessarily hold for other projections (Wasserstein, linear as discussed in Proposition 2.3 and Proposition B.1).

---

> > ### Author Response · Authors · 2021-11-11
> > **Response to Reviewer nH3D (Part 2)**
> >
> > ### 4. Regarding taking expectation in SSD divergence definition:
> >
> > We start our derivation of SSD divergence from the Hausdorff distance. Since the Hausdorff distance is effectively only calculating the greatest distance between a point and a set from two distributions, it is an ineffective training objective, as it only provides a signal to a single point. Taking the expectation solves this problem, as it provides a stronger signal by taking into account all the points in the support difference: $\operatorname{supp}(p) \triangle \operatorname{supp}(q)$.
> >
> > We formulate SSD divergence as an expectation in equation (1) for general (potentially continuous) distributions. Note that for discrete (empirical) distributions $p(x) = \frac{1}{N} \sum \limits_{i=1}^N \delta (x - a_i)$, $q(x) = \frac{1}{M} \sum \limits_{i=1}^M \delta (x - b_i)$, the expectation in the expression for SSD divergence naturally reduces to average: $\mathcal{D}_{\triangle}(p, q) = \mathbb{E}_p [d(x, \operatorname{supp}(q)] + \mathbb{E}_q [d(x, \operatorname{supp}(p)]$
> >
> > $= \frac{1}{N} \sum\limits_{i=1}^N \min\limits_{1 \leq j \leq M} d(a_i, b_j) + \frac{1}{M} \sum\limits_{i=1}^M \min\limits_{1 \leq j \leq N} d(b_i, a_j)$.
> > In practice, the proposed ASA algorithm uses this version of the averaged distance over samples in the 1D discriminator output space (equation 5).

---

> > > ### Author Response · Authors · 2021-11-11
> > > **Response to Reviewer nH3D (Part 3)**
> > >
> > > ### 5. Regarding comparison with optimal transport based baselines in experiments:
> > >
> > > We provide additional experimental results comparing our method with OT-based methods for domain adaptation. We implement two OT-based methods which we describe below.
> > > 1. The first method is a variant of the max-sliced Wasserstein distance (which was proposed for GAN training by Deshpande et al., 2019) for domain adaptation. In the table below we refer to this method as DANN-OT. In our implementation DANN-OT minimizes the Wasserstein distance between the pushforward distributions $g^*_{\sharp} p$, $g^*_{\sharp} q$ induced by the optimal log-loss discriminator $g^*$. As discussed in Section 4.2 (paragraph “Distribution alignment”) the computation of the Wasserstein distance between 1D distributions can be implemented efficiently via sorting.
> > > 2. The second method is an OT-based variant of DANN which uses a dual Wasserstein discriminator instead of the log-loss discriminator. In the table below we refer to this method as DANN-WGP. This method minimizes the Wasserstein distance in its dual Kantorovich form. We train the discriminator with the Wasserstein dual objective and a gradient penalty proposed to enforce
> > > Lipshitz-norm constraint (Gulrajani et al., 2017).
> > >
> > > We present the evaluation results of the OT-based methods on STL->CIFAR domain adaptation task in the table below. The evaluation metrics are the same as in Table 1 (in each cell the first set of numbers show the statistics of the class-average target accuracy, and the second set of numbers shows the statistics of the worst target class accuracy).
> > >
> > > | Algorithm   | $\alpha = 0.0$                                    | $\alpha = 1.0$                                | $\alpha = 1.5$                                         | $\alpha = 2.0$                                    |
> > > |:------------|:--------------------------------------------------|:--------------------------------------------------|:--------------------------------------------------|:--------------------------------------------------|
> > > | No DA                 | $ 69.9_{69.8}^{70.0} $ $ 49.8_{45.3}^{50.6} $ | $ 68.8_{68.3}^{69.3} $ $ 47.2_{45.3}^{48.2} $ | $ 66.8_{66.4}^{67.2} $ $ 46.0_{45.8}^{47.0} $ | $ 65.8_{64.8}^{66.7} $ $ 43.7_{41.6}^{44.6} $ |
> > > | DANN (log-loss)       | $ 75.3_{74.9}^{75.4} $ $ 54.6_{54.2}^{56.6} $ | $ 69.9_{68.6}^{70.1} $ $ 44.8_{40.7}^{45.1} $ | $ 64.9_{63.7}^{67.1} $ $ 34.9_{33.9}^{36.8} $ | $ 63.3_{57.4}^{64.8} $ $ 27.0_{21.2}^{28.5} $ |
> > > | DANN-OT             | $ 76.0_{75.8}^{76.0} $ $ 55.2_{54.3}^{55.5} $ | $ 67.7_{67.1}^{68.9} $ $ 43.0_{36.5}^{43.7} $ | $ 64.5_{60.9}^{65.1} $ $ 34.4_{29.3}^{34.6} $ | $ 61.3_{54.4}^{62.0} $ $ 24.3_{23.2}^{25.5} $ |
> > > | DANN-WGP | $ 74.8_{74.7}^{75.1} $ $ 53.5_{53.3}^{54.4} $ | $ 67.7_{65.3}^{67.9} $ $ 38.6_{34.4}^{41.0} $ | $ 63.3_{57.1}^{63.4} $ $ 27.0_{26.3}^{32.4} $ | $ 59.0_{54.3}^{61.8} $ $ 21.9_{18.6}^{22.5} $ |
> > > | sDANN-4               | $ 71.8_{71.7}^{72.1} $ $ 52.1_{52.1}^{52.8} $ | $ 71.1_{70.4}^{71.7} $ $ 49.9_{48.1}^{51.8} $ | $ 69.4_{68.7}^{70.0} $ $ 48.6_{43.5}^{49.0} $ | $ 66.4_{66.2}^{67.9} $ $ 39.0_{33.6}^{47.1} $ |
> > > | ASA-sq                | $ 71.7_{71.7}^{71.9} $ $ 52.9_{46.7}^{53.4} $ | $ 70.7_{70.4}^{71.0} $ $ 51.6_{46.8}^{52.7} $ | $ 69.2_{69.2}^{69.3} $ $ 45.6_{43.3}^{52.0} $ | $ 68.1_{67.2}^{68.2} $ $ 44.7_{39.8}^{45.9} $ |
> > > | ASA-abs            | $ 71.6_{71.2}^{71.7} $ $ 49.0_{48.4}^{53.5} $ | $ 70.9_{70.8}^{71.0} $ $ 49.2_{47.3}^{50.0} $ | $ 69.6_{69.6}^{69.9} $ $ 43.2_{42.1}^{49.5} $ | $ 67.8_{66.6}^{68.2} $ $ 40.9_{35.4}^{49.0} $ |
> > >
> > > Note that the OT-based methods aim to enforce distribution alignment constraints. We observe that the OT-based methods follow the same trend as DANN: they deliver improved accuracy compared to No DA in the balanced setting, but suffer in the imbalanced settings ($\alpha > 0$) due to their distribution alignment nature.
> > >
> > >
> > > We would also like to make a comment on OT-based relaxed distribution alignment. Wu et al. (2019b) proposed the WDANN-$\beta$ method which minimizes the dual form of the asymmetrically-relaxed Wasserstein distance. However, Wu et al. observed that sDANN-$\beta$ outperforms WDANN-$\beta$ in experiments. As a result, we use sDANN-$\beta$ as a relaxed distribution alignment baseline in our experiments.
> > >
> > > We are happy to answer any further questions regarding the paper.

---

> ### Comment · Reviewer_nH3D · 2021-11-29
> **Thanks for the response**
>
> Thank you for the response.
>
> As in the revised version, SSD is essentially the same as Chamfer distance (at least its discrete version, and perhaps the contribution is the "continuous version of Chamfer distance"?). I think the authors should place the proposed SSD in the literature context, so it can highlight the contributions of the paper. But again, I am confused when the authors focus on 1d projection (which helps the computation) in case the authors would like to advocate the "continuous version" of Chamfer (In my understanding, one may not compute the continuous version directly but only can estimate it even in the 1d case, e.g., Gaussian distribution where supports are "unbounded").
>
> In terms of concept and applications, I think the authors did a good job.
>
> It seems better in case the authors should describe the relation SSD and Chamfer more clearly and sooner (currently, it is placed at the end of Section 6) since they are essentially the same (in the discrete case) in the updated version.

---

> > ### Author Response · Authors · 2021-11-29
> > **Response to the follow-up comment**
> >
> > We thank the reviewer for their follow-up comment, and we appreciate that the reviewer acknowledges our main theoretical and empirical contributions besides the SSD divergence.
> >
> > We cannot upload a revised version of the text during this final stage of discussion, but we will make edits according to your suggestions. In particular, we will discuss the relationship between SSD and Chamfer/RWMD when we introduce SSD in Section 2.1.
> >
> > The continuous version of Chamfer, SSD divergence, enables us to work with densities, which is required for our theoretical analysis in Section 2.2 and 4.1 (these parts focus on the validity of 1D projection via the optimal log-loss discriminator). One can compute the SSD divergence between continuous distributions if given direct access to the distributions: in the case of two Gaussians, since they have full supports, the SSD divergence equals 0. Distributions in practice do not have full supports. We estimate the SSD divergence in 1D via i.i.d. samples. We assume for theoretical reasons that the samples come from densities so that we can talk about projecting distributions to 1D and estimate the associated supports. This is the same setup as many existing formulations and frameworks. For example, in the case of GANs, the global optimality is proven by assuming the distributions have densities while the actual algorithms operate with i.i.d. samples.

---

> > ### Author Response · Authors · 2021-12-01
> > **A note to Reviewer nH3D**
> >
> > We thank the reviewer again for their input and for the opportunity given to us to clarify our contributions. We feel that we have adequately addressed all the concerns expressed about the paper. We respectfully ask that the reviewer revisits their initial assessment to better reflect post discussion evaluation.

---

### Official Review · Reviewer_BDME · 2021-11-02

**Correctness:** 3
**Technical Novelty And Significance:** 4
**Empirical Novelty And Significance:** 3
**Recommendation:** 8
**Confidence:** 4

**Main Review:**

**Strengths:**
- Proposes a novel support (pseudo-)divergence (not a true divergence because D = 0 means the supports match but possibly the distributions do not.) to measure the difference in supports between distributions.  This is an extension of the Hausdorff distance.

- Theoretically proves that alignment of the support of the optimal discriminator implies alignment of the distributions and gives several insights about alternatives.

- Place the support alignment goal and objectives within related relaxed and full alignment objectives.

- Extensive empirical experiments under the label shift setting.

**Weakness:**
- My main concern is that one solution to support alignment is distribution alignment.  And more importantly, the algorithm (as shown by the mini-batch part) seems to naturally collapse to distribution alignment (as evidenced by the mini-batch issue).  However, the label shift application suggests that a better solution is to align the supports but NOT align the whole distribution (even though that is one solution). Thus, it seems an explicit regularization or something should be added to the objective rather than an ad-hoc history buffer.  One suggestion might be a transportation cost regularization to the transformation (see below).

- Another issue is that the support alignment method is only evaluated using the extrinsic task of domain adaptation.  The paper does not include results on evaluating the support alignment explicitly.  Even a simulated experiment may help to evaluate the algorithm with respect to support alignment directly.

**Minor Weaknesses:**
- The history buffer idea seems to be entirely heuristic.  Can you explain why this might be important theoretically? Or why without it there may be a collapse to distribution alignment?  My intuition is that the objective will naturally move towards shrinking these values towards 1/2 as mentioned below in comments but maybe there is something else going on.

- The idea of "constraints" being an issue for distribution alignment seems a little odd (as in the first figure).  The solution seems to be to increase the model capacity rather than align the support of the distribution. This seems to be a fairly weak high-level motivation for support alignment (at least in the introduction). The shift in label distributions is a stronger motivation in my view and should be the primary example.

- Proposition 2.2 is a bit confusing to understand.  Maybe a little illustration would be helpful to show the densities of $\tau_p$ and $\tau_q$ and the resulting ratio of densities for $\tau$ along with the $x$ and $x'$ that was mentioned in the explanation.

- The paper should make a clearer distinction between "continuous" support alignment (which seems to be the overall benefit) and discrete support alignment (which seems to be the case in the algorithm which uses samples).


**Other comments or questions**
- It is interesting that full distribution alignment corresponds to the case when the optimal discriminator is just 1/2.  Thus, it seems that concentrating all mass near 0.5 is really distribution alignment, while "spreading" out the mass of the $\tau$ predicted probability distributions (except for 0 and 1) means that the supports are aligned.  Is this a correct intuition/interpretation?  Is this also why mini-batch training actually finds this solution (as it is the most stable solution maybe)---i.e., we can always improve the objective by shrinking towards 1/2, which is the full distribution alignment case?

- If the above interpretation is correct, then it would seem that you could actually enforce support alignment but NOT distribution alignment by requiring $\tau$ to have high entropy.

- It seems that another way to avoid this "collapsing" phenomena would be to add a transportation cost term to the transformations.  Thus, we would seek to move the points as little as possible to align the support of the distributions. This would also be much simpler and more grounded than the history idea. Have you considered this idea?

- Proposition 4.3 seems quite strong.  Is statement 1 related to max sliced wasserstein distance (i.e., joint Wasserstein is 0 if and only if max sliced Wasserstein is 0)?

- The result tables should be shown after the description of the methods, metrics, etc.

- I might suggest rewriting with label shift as the core issue as this seems to be the most realisitic example and iis strongly supported by your experiments.  Other settings are not demonstrated by your experiments and are not well-motivated.



**Summary Of The Paper:**

Motivated by label shift between source and target domains, the paper proposes a novel support alignment task with a novel support divergence.  They provide theoretical results for these and build up theory that leads to a simple adversarial algorithm for support alignment using a discriminator and nearest neighbor algorithms in 1D.
The paper provides extensive empirical results comparing to common domain adaptation methods.


**Summary Of The Review:**

Overall, the paper presents novel ideas and tasks and places the work well within existing literature.  This could have significant impact on unsupervised domain alignment and more generally on thinking about alignment problems. The main weakness is the potential for the proposed algorithm to collapse to standard distribution alignment and the lack of explicit evaluation of support alignment.

---

> ### Author Response · Authors · 2021-11-17
> **Response to Reviewer BDME (Part 1)**
>
> We thank the reviewer for their detailed review and insightful comments. We address the reviewer’s concerns below.
>
> > My main concern is that one solution to support alignment is distribution alignment. And more importantly, the algorithm (as shown by the mini-batch part) seems to naturally collapse to distribution alignment (as evidenced by the mini-batch issue). However, the label shift application suggests that a better solution is to align the supports but NOT align the whole distribution (even though that is one solution). Thus, it seems an explicit regularization or something should be added to the objective rather than an ad-hoc history buffer. One suggestion might be a transportation cost regularization to the transformation (see below).
>
> > The history buffer idea seems to be entirely heuristic. Can you explain why this might be important theoretically? Or why without it there may be a collapse to distribution alignment? My intuition is that the objective will naturally move towards shrinking these values towards 1/2 as mentioned below in comments but maybe there is something else going on.
>
> Support alignment is a more permissive alignment criterion than distribution alignment. Indeed, distribution alignment implies support alignment but aligning only the support does not mean that distributions are otherwise aligned. We can obtain support alignment by optimizing SSD divergence on the dataset level. This is not, however, feasible computationally (iteration cost). Minimizing SSD divergence per minibatch (rather than over the full dataset in each iteration) changes the overall support alignment criterion towards distribution alignment. The history buffer is used to explicitly counter and control this effect.
>
> Without history buffers, we are left with minimizing equation (5) with only the current batch samples. Theoretically, instead of minimizing SSD, we are aligning the supports of two randomly sampled mini-batches. In the paper, we talked about how support alignment for all possible pairs of mini-batches is a much stricter constraint, since support alignment on all possible pairs of mini-batches implies support alignment on the whole dataset but not vice versa.  A more intuitive explanation that relates mini-batch support alignment to distribution alignment is: the only case, where every pair of randomly sampled mini-batches share the same support in the optimal discriminator output space, is when both source and target distributions are concentrated at a single point ($\frac{1}{2}$), implying distribution alignment.
>
> Following this reasoning, we propose using history buffers to approximate the whole dataset supports in 1D discriminator output space with very light computation overhead. Empirically, we show the impact of the history buffers on the distribution-vs-support alignment in the table presented in Part 2 below. The table shows that the size of the history buffer clearly relates to how much distributions are permitted to misalign while being support aligned.

---

> > ### Author Response · Authors · 2021-11-17
> > **Response to Reviewer BDME (Part 3)**
> >
> > > The idea of "constraints" being an issue for distribution alignment seems a little odd (as in the first figure). The solution seems to be to increase the model capacity rather than align the support of the distribution. This seems to be a fairly weak high-level motivation for support alignment (at least in the introduction). The shift in label distributions is a stronger motivation in my view and should be the primary example.
> >
> > When we say “constraints” about distribution alignment, we mean that it is a stricter requirement compared to support alignment, as alignment in distributions implies alignment in supports but not vice versa. However, the minimizer for a distribution alignment objective does not necessarily minimize a support alignment objective when there is an additional objective/constraint to satisfy, and the motivating figure/example is designed to demonstrate that. The focus is not on the particular constraint we used (the model capacity) for the example, but rather the qualitative distinction between two minimizers when distribution alignment is not achievable (the reason could be model capacity in the example or that we want to achieve good adaptation performance in our experimental setup). We choose to motivate the paper this way because we believe support alignment itself is a well-defined and challenging problem, which should not be tied with a particular use case of domain adaptation.
> >
> >
> >
> >
> > > It is interesting that full distribution alignment corresponds to the case when the optimal discriminator is just 1/2. Thus, it seems that concentrating all mass near 0.5 is really distribution alignment, while "spreading" out the mass of the  predicted probability distributions (except for 0 and 1) means that the supports are aligned. Is this a correct intuition/interpretation? Is this also why mini-batch training actually finds this solution (as it is the most stable solution maybe)---i.e., we can always improve the objective by shrinking towards 1/2, which is the full distribution alignment case?
> >
> > > If the above interpretation is correct, then it would seem that you could actually enforce support alignment but NOT distribution alignment by requiring  to have high entropy.
> >
> > > It seems that another way to avoid this "collapsing" phenomena would be to add a transportation cost term to the transformations. Thus, we would seek to move the points as little as possible to align the support of the distributions. This would also be much simpler and more grounded than the history idea. Have you considered this idea?
> >
> > The understanding is mostly correct, but we would like to note some details about it. Indeed, full distribution alignment is achieved when the range of optimal discriminator output values shrinks to just a single point, ½. However, support alignment does not correspond to expanding the range. Recall that distribution alignment is an acceptable solution to support alignment, so generally having just a shared ½ as output is fine for support alignment. In theory, the larger the output range is, the greater the maximum density ratio is ($\frac{p}{p+q}$ and $\frac{q}{p+q}$). This understanding is a direct result of Proposition 2.2. Explicitly making the output range wider is essentially rejecting solutions with lower maximum density ratios ($\frac{p}{p+q}$ and $\frac{q}{p+q}$ are bounded by some value everywhere), including distribution alignment. In this paper, we focus on presenting a general way to align just distribution supports, which means we do not impose any structure on the distributions other than support alignment.
> >
> > If our proposed ASA method works on the whole dataset without utilizing mini-batches, then it essentially moves the points as little as possible to align the support of the distributions. Intuitively, the optimal discriminator gives level sets of the ratio of densities ($\frac{p}{p+q}$). One particular point only receives a (gradient) signal to move in a direction orthogonal to the level sets, either “inward” or “outward”, and it stops receiving signals from the alignment objective once there is a point from the other distribution on the same level set. Effectively every point moves as little as possible to align supports when co-training with the discriminator. However, as we have described in the previous parts of the response, the challenge is on how accurate the estimation of the whole dataset support is, and an inaccurate estimation would lead to inaccurate movement signals.
> >
> >
> > > Proposition 4.3 seems quite strong. Is statement 1 related to max sliced wasserstein distance (i.e., joint Wasserstein is 0 if and only if max sliced Wasserstein is 0)?
> >
> > Yes, it is related. However, note that for max-sliced Wasserstein distance, the transformation/projection is linear, whereas we consider the transformation induced by a learned nonlinear discriminator. This distinction requires different proofs.
> >
> > We are happy to answer any further questions regarding the paper.

---

> > ### Author Response · Authors · 2021-11-17
> > **Response to Reviewer BDME (Part 2)**
> >
> > > Another issue is that the support alignment method is only evaluated using the extrinsic task of domain adaptation. The paper does not include results on evaluating the support alignment explicitly. Even a simulated experiment may help to evaluate the algorithm with respect to support alignment directly.
> >
> > To explicitly evaluate our ASA method (especially with the usage of history buffers) in terms of support alignment, we present comparisons between embedding spaces resulting from different alignment objectives (no alignment, distribution alignment, and support alignment) on USPS->MNIST task. We investigate a simpler version of the original problem presented in Section 5, in particular, a 3-class classification using a feature extractor with a 2D feature representation (embedding space). We select a subset of 3 classes (digits 3, 5, 9) from both source and target datasets and, following the domain adaptation setup used in the paper, we subsample the datasets so that the label distribution on source is balanced and the target label distribution follows power law with $\alpha=1.5$.  As for the evaluation metrics that compare the behaviors of different methods, we choose Wasserstein-1 distance to measure the degree of distribution alignment and SSD divergence to measure the degree of support alignment, both in the original embedding space without the learned discriminator transformation to 1D. In order to have comparable distances between embedding spaces learned by different methods, we normalize the embeddings before computing the distribution/support distances: we apply a global affine transformation for each embedding space by 1) centering the embeddings so that their average is 0; 2) re-scaling the embeddings so that their average norm is 1. Similarly to the experiments presented in the paper we make 5 runs with different seeds and report the median (the main number), 25 (subscript), and 75 (superscript) percentiles.
> >
> >
> > | Algorithm         | Hisotry size | Wasserstein-1          | SSD                    |
> > |:------------------|:------------:|:-----------------------|:-----------------------|
> > | No DA             | --           | $ 0.78_{0.75}^{0.84} $ | $ 0.10_{0.10}^{0.10} $ |
> > | DANN              | --           | $ 0.07_{0.06}^{0.08} $ | $ 0.02_{0.02}^{0.02} $ |
> > | ASA-abs           | 0            | $ 0.23_{0.22}^{0.47} $ | $ 0.03_{0.03}^{0.03} $ |
> > | ASA-abs           | 100          | $ 0.54_{0.36}^{0.56} $ | $ 0.03_{0.03}^{0.03} $ |
> > | ASA-abs           | 500          | $ 0.59_{0.55}^{0.64} $ | $ 0.03_{0.03}^{0.03} $ |
> > | ASA-abs           | 1000         | $ 0.59_{0.55}^{0.62} $ | $ 0.03_{0.03}^{0.03} $ |
> > | ASA-abs           | 2000         | $ 0.62_{0.58}^{0.66} $ | $ 0.03_{0.03}^{0.03} $ |
> > | ASA-abs           | 5000         | $ 0.64_{0.63}^{0.67} $ | $ 0.04_{0.04}^{0.04} $ |
> >
> >
> >
> > The above table shows that, compared to no alignment and distribution alignment (DANN) methods, ASA aligns the supports without necessarily aligning the distributions (in this imbalanced case, distribution alignment implies low adaptation accuracy).
> >
> > We will include the results of this experiment in the revised version of the paper. In addition to the above table, we will also include figures visualizing the 2D embedding spaces learned by different alignment methods, so that the difference between no alignment, distribution alignment, and support alignment can be assessed visually. From our current visualizations, we observe that both DANN and ASA achieve support alignment compared to No DA. While DANN enforces a stricter distribution alignment constraint and thus places some of the target embeddings into regions corresponding to wrong classes, ASA (with history buffers) does not enforce distribution alignment and maintains good class correspondence across the source and target embeddings.
> >
> > > Proposition 2.2 is a bit confusing to understand. Maybe a little illustration would be helpful to show the densities of $\tau_p$ and $\tau_q$ and the resulting ratio of densities for $\tau$ along with the $x$ and $x'$ that was mentioned in the explanation.
> >
> > We will add an illustration intuitively explaining the results. Essentially, if the optimal discriminator $f^*$ maps $x\in\mathcal{X}$ to $a\in[0,1]$, i.e. $f^*(x)=a$, $a$ directly corresponds to the ratio of densities in not only the original space $a=\frac{p(x)}{p(x)+q(x)}$, but also the discriminator output space $a=\frac{\tau_p(a)}{\tau_p(a)+\tau_q(a)}$, which leads to the fact that $f^*(x)=a=\frac{p(x)}{p(x)+q(x)}=\frac{\tau_p(a)}{\tau_p(a)+\tau_q(a)}$.
> >
> > > The paper should make a clearer distinction between "continuous" support alignment (which seems to be the overall benefit) and discrete support alignment (which seems to be the case in the algorithm which uses samples).
> >
> > We will add a dedicated paragraph on the differences between discrete and continuous support alignment.

---

> ### Comment · Reviewer_BDME · 2021-11-24
> **Post-rebuttal response**
>
> I thank the reviewers for their thoughtful responses.  I definitely appreciated the additional explanations and in particular the new Figure A.1 and Figure 2.  I did have to stare at Figure A.1 for a while but am beginning to understand better.
>
> As a followup on Figure A.1, can I think of the ratio as the same while the absolute value of the pushforward distribution is related to how often the distributions achieve that ratio?  Is this related to the length of the level sets or more related to the original mass on those level sets?  It seems like it can be thought of as a line integral on the level set of the optimal classifier.
>
> I also appreciated your explanation about how support alignment isn't about making the pushforward have higher variance but rather moving the points as little as possible to achieve alignment.  I am still a bit interested in the fact that full alignment is one solution, but it seems you really want a solution on the edge of the feasible set.  Would it be reasonable to regularize the pushforward distribution towards the original pushforward distribution (before alignment)?  This may help encourage a more unique solution rather than just hoping that the algorithm itself (using history buffers) finds a unique solution "on the edge" of the feasible set.  Or, it might be best to add a transportation cost to the objective to explicitly ensure a non-fully aligned solution.
>
> Regardless, I still appreciate this paper and recommend it's acceptance.

---

> > ### Author Response · Authors · 2021-11-30
> > **Response to the follow-up comment**
> >
> > Thank you for the follow-up comment. We are glad that we are able to improve the presentation of our results in the paper.
> >
> > Regarding Figure A.1. Indeed, the value of the pushforward density $\tau_p(a)$ shows how often $f^*(x)$ takes the value of $a$ with $x \sim p$. Formally, $\tau_p(a)$ is related to the probability mass of the level sets of $f^*$. In our proof of Proposition 2.2 we represent the pushforward density as
> >
> > $\tau_p(a) = \lim_{\varepsilon \downarrow 0}
> >     \frac{\mathbb{P}_p\left(
> >     \{x \mid a - \varepsilon < f^*(x) < a + \varepsilon \}
> >     \right)}{2\varepsilon}$,
> >
> > where $\mathbb{P}_p \left( \{ x \mid a - \varepsilon < f^*(x) < a + \varepsilon\}\right)$ can be intepreted as the probability mass of $\varepsilon$-tolerance level set of $f^*$.
> >
> > There are many different solutions that satisfy the support alignment constraint and by including additional regularization objectives one could steer the method towards solutions with certain characteristics desired in a given problem. For instance, in Section D.3 we show that our implementation of support alignment benefits from having additional regularization objectives (conditional entropy or VAT-loss) in the context of domain adaptation. We appreciate your suggestions and we believe that the design of such regularization techniques is a promising direction for future work.

---

### Official Review · Reviewer_ukYK · 2021-11-04

**Correctness:** 4
**Technical Novelty And Significance:** 3
**Empirical Novelty And Significance:** 3
**Recommendation:** 8
**Confidence:** 3

**Main Review:**

Strengths:
Overall a great paper. Very easy to follow, the algorithm seem theoretically grounded and simple and very well motivated. It's a very important domain, and the story is pretty clear in that regard. Uses well-known tools (e.g., GAN discriminator), so should be easy to implement. Experiments are well-motivated and support the approach.

Weaknesses (not really) / comments:
One thing that would be good in this work is to see this applied to unsupervised domain translation (e.g., converting SVHN to MNIST). There are a number of works that attempt to address this problem (e.g., Binkowski 2019), but they come with a great deal of complexity. Have you tried this approach?

I feel like d(.,.) could have been clarified better earlier, at least to provide some examples to avoid the reader wondering about what it was until the experiment section.

Binkowski 2019: Batch weight for domain adaptation with mass shift

**Summary Of The Paper:**

This paper introduces a method of aligning the support of one distribution to another. This is important as for domain adaptation problems, distributional alignment can come with a lot of problems (such as moving the mass of one class to another when there are differences in representation of different classes).

**Summary Of The Review:**

The paper is very clear, well motivated, the approach is simple, and problem is important.

---

> ### Author Response · Authors · 2021-11-17
> **Response to Reviewer ukYK**
>
> We thank the reviewer for their comments and interesting suggestions. We provide our responses below.
>
> **Regarding applying support alignment to unsupervised domain translation**
>
>   We thank the reviewer for bringing up this direction. While we have not yet tried this setup with support alignment, we agree it is an interesting problem for which our method should be suitable. The potential application in this direction also agrees with our vision: support alignment is a generally interesting and useful problem to consider, and should not be tied with a particular use case of domain adaptation. We will add a discussion on this future direction to our paper.
>
> **Regarding clarifying $d(\cdot, \cdot)$ earlier**
>
>   We will add examples of commonly used metric $d(\cdot, \cdot)$ when we introduce it in the Notation paragraph, e.g. Euclidean distance.
>
>
> We are happy to answer any further questions regarding the paper.

---

### Author Response · Authors · 2021-11-21
**Updated version of the paper**

We thank all reviewers for the helpful and constructive feedback on our paper.

We have updated the paper. In the updated version we include clarifications on the reviewers’ comments as well as additional experimental results we presented in our responses to individual reviews.

We made the following changes:
* (following reviewer **nH3D**’s comments) included discussion of the relationship between SSD divergence and Chamfer distance in the related work section (paragraph “Relaxed distribution alignment”);
* (following reviewer **nH3D**’s comments) included results of comparison with OT-based domain adaptation on STL->CIFAR dataset (Appendix D.3);
* (following comments of reviewers **nH3D** and **ukYK**) included references to ArXiv1612.00603, ArXiv2102.04014, and Binkowski, et. al 2019.
* (following comments of reviewers **nH3D** and **BDME**) clarified the distinction between continuous and discrete distributions as well as expectation/average at the beginning of Section 3;
* (following reviewers **FsUp**’s comments) included the results of ablation studies (effect of the conditional entropy loss, effect of alignment weight) in Appendix D.3; also added the expression for the conditional entropy loss in Appendix D.2;
* (following comments of reviewers **FsUp** and **ukYK**) we added “Conclusion and Future Work” section (Section 7) summarizing the results of the paper and discussing potential directions for future work (domain translation, partial DA);
* (following reviewers **FsUp**’s comments) mentioned the use of single discriminator update in practice (footnote in the first paragraph of Section 3);
* (following reviewers **FsUp**’s comments) updated the proof of Proposition 2.3 by providing a counter-example with distributions having densities;
* (following reviewer’s **BDME**’s comments) included results of direct evaluation of support alignment in 2D embedding space in Section 5 (paragraph “Illustrative example”, Figure 2) and Appendix D.5 (Table D.5). Note that we moved the “History size effect” paragraph from Section 5 to Appendix D.5 and replaced it with the “Illustrative example paragraph”. We also included Figure 2 which shows a visualization of the learned 2D embedding spaces where support alignment can be assessed qualitatively;
* (following reviewer’s **BDME**’s comments) provided an illustration of the statement of Proposition 2.2 in Figure A.1;
* (following reviewer’s **BDME**’s comments) moved the result tables in Section 5 so that they appear after the description of the methods and evaluation metrics;
* (following reviewer **ukYK**’s comments) mentioned an example of a distance d(.,.) in the “Notation” part of  Section 2;
* fixed typos and generally revised the text improving the clarity of the presentation.

---

### Decision · Program_Chairs · 2022-01-20

**Decision:**

Accept (Spotlight)

**Comment:**

Thanks for your submission to ICLR.

Three of the four reviewers are ultimately (particularly after discussion) very enthusiastic about the paper, and feel that their concerns have been adequately addressed.  The fourth reviewer has not updated his/her score but has indicated that their concerns were at least somewhat addressed.  I took a look at their review and agree that the authors have addressed these concerns sufficiently.  I am happy to recommend this paper for acceptance at this point.  Note that I really appreciate the time and effort that the authors went into adding additional results and clarifications for the reviewers.